# Coding strategies in the otolith system differ for translational head motion vs. static orientation relative to gravity

Mohsen Jamali[1], Jerome Carriot[2], Maurice J Chacron[2], Kathleen E Cullen[3]*

[1]Department of Neurosurgery, Harvard Medical School, Massachusetts General Hospital, Boston, United States; [2]Department of Physiology, McGill University, Montreal, Canada; [3]Department of Biomedical Engineering, Johns Hopkins University, Baltimore, United States

**Abstract** The detection of gravito-inertial forces by the otolith system is essential for our sense of balance and accurate perception. To date, however, how this system encodes the self-motion stimuli that are experienced during everyday activities remains unknown. Here, we addressed this fundamental question directly by recording from single otolith afferents in monkeys during naturalistic translational self-motion and changes in static head orientation. Otolith afferents with higher intrinsic variability transmitted more information overall about translational self-motion than their regular counterparts, owing to stronger nonlinearities that enabled precise spike timing including phase locking. By contrast, more regular afferents better discriminated between different static head orientations relative to gravity. Using computational methods, we further demonstrated that coupled increases in intrinsic variability and sensitivity accounted for the observed functional differences between afferent classes. Together, our results indicate that irregular and regular otolith afferents use different strategies to encode naturalistic self-motion and static head orientation relative to gravity.

DOI: https://doi.org/10.7554/eLife.45573.001

*For correspondence:
kathleen.cullen@jhu.edu

**Competing interests:** The authors declare that no competing interests exist.

## Introduction

The otolith system provides vital information about linear head acceleration in three dimensions (i.e., gravito-inertial forces) and the head's orientation relative to gravity. This essential system is required for control of balance and posture, as well as for our perception of self-motion and spatial orientation. Indeed, during natural everyday activities, the head not only moves with considerable linear acceleration (frequencies up to 20 Hz and amplitudes up to 8G; see *Carriot et al., 2014* for human data and *Carriot et al., 2017* for monkey and mouse data), but also frequently remains stationary during large periods of time (*Carriot et al., 2014*; *Carriot et al., 2017*). This observation raises the fundamental question of how the otolith system provides estimates of both dynamic head motion and static orientation relative to gravity.

Otolith afferent responses have been traditionally characterized using artificial self-motion stimuli such as sinusoids (reviewed in *Goldberg, 2000*, *Cullen, 2012*, *Yu et al., 2012*, and *Jamali et al., 2013*) or single Gaussian-like trajectories (*Yu et al., 2015*; *Laurens et al., 2017*). In the absence of stimulation, otolith afferents display differences in the variability of their resting discharge and can be classified as regular or irregular. Regular and irregular afferents further display different morphological and physiological features (reviewed in *Goldberg, 2000*, *Cullen, 2012* and *Eatock and Songer, 2011*). In addition, a number of studies have recorded otolith afferent responses to sound and/or bone vibrations that have frequency content that is an order of magnitude above that of the self-motion experienced during everyday activities (*Young et al., 1977*; *Curthoys et al., 2019*;

*Curthoys et al., 2016*; *Curthoys et al., 2017*). Interestingly, irregular otolith afferents show greater phase locking (i.e., firing only during a specific phase range of the sinusoidal stimulus) at higher frequencies (i.e., 200–3,000 Hz). However, how the otolith afferents respond to the naturalistic self-motion stimuli experienced during everyday activities remains unknown. Specifically, studies have not revealed the role played by heterogeneity in the resting discharge variability of otolith afferents in the coding strategies employed by these neurons.

Accordingly, here we directly addressed how otolith afferents encode the gravito-inertial forces experienced during everyday activities. We found that irregular afferents displayed strong response nonlinearities to naturalistic self-motion and transmitted information via both changes in firing rate and precise spike timing. By contrast, regular afferents primarily encoded naturalistic self-motion through changes in firing rate. Mathematical modeling revealed that the increased sensitivity and variability of irregular afferents could account for functional differences between regular and irregular otolith afferents. Our model further predicted that irregular afferents should display temporally precise phase locking in response to sinusoidal stimulation within the behaviorally relevant physiological range. Further experiments using sinusoidal stimuli confirmed this prediction. Finally, we found that regular afferents outperformed their irregular counterparts in their ability to discriminate between different static head orientations. Taken together, our results establish that regular and irregular otolith afferents use different coding strategies in order to provide estimates of both dynamic head motion and static orientation relative to gravity.

## Results

We recorded the activities of primary otolith afferents in two alert macaque monkeys (*Figure 1A*). Resting discharge variability was quantified from interspike interval histograms (upper left and right panels of *Figure 1B*) using a normalized coefficient of variation CV* (see 'Materials and methods'). We found that the distribution of CV* values for our dataset was bimodal (*Figure 1B*, p=0.04, Hartigan's dip test), consistent with previous studies (reviewed in *Goldberg, 2000*). Our afferent dataset thus comprised N = 18 regular and N = 17 irregular afferents. We then recorded the activities of primary otolith afferents under stimulation. Our experimental protocol comprised: 1) translational self-motion stimuli with frequencies spanning the natural range (0–15 Hz) mimicking natural movements (*Carriot et al., 2017*) (see 'Materials and methods') and; 2) different head orientations relative to gravity obtained by statically re-aligning the animal's whole body. We henceforth refer to these two stimuli 'naturalistic translational self-motion' and 'changes in head orientation relative to gravity', respectively. We note that we applied the latter in order to investigate otolith afferent coding for frequencies <0.1 Hz, as we could only reliably estimate quantities for frequencies ≥0.1 Hz for naturalistic translational self-motion (see 'Materials and methods').

### Irregular afferents display stronger response nonlinearities and transmit more information overall about naturalistic translational self-motion stimuli than their regular counterparts

*Figure 1C* shows the responses of typical irregular (top) and regular (bottom) otolith afferents to naturalistic translational self-motion stimuli. Notably, changes in firing rate were greater for the irregular afferent (*Figure 1C*, compare red and blue curves). Further analysis of neuronal responses across our dataset revealed that irregular afferents displayed greater sensitivity to the stimulus than their regular counterparts (*Figure 1D* and *Figure 1D*, left inset). Using linear systems identification techniques, we generated an estimate of the time-dependent firing rate in response to stimulation (see 'Materials and methods'). Overall, there was good agreement between this prediction and the actual firing rate for both regular and irregular afferents (compare blue and gray for regular afferents and red and gray for irregular afferents in *Figure 1C*). In order to quantify the component of the response that could not be explained by our linear model, we computed the residual (i.e., the difference between our estimate and the actual firing rate). Overall, this residual was distributed over a greater range of values for irregular afferents than for their regular counterparts (compare dashed red and blue curves in *Figure 1C*). Further analysis of the residual power spectrum revealed nearly constant power, which was higher by over an order of magnitude for irregular afferents (*Figure 1D*, right inset, compare red and blue curves).

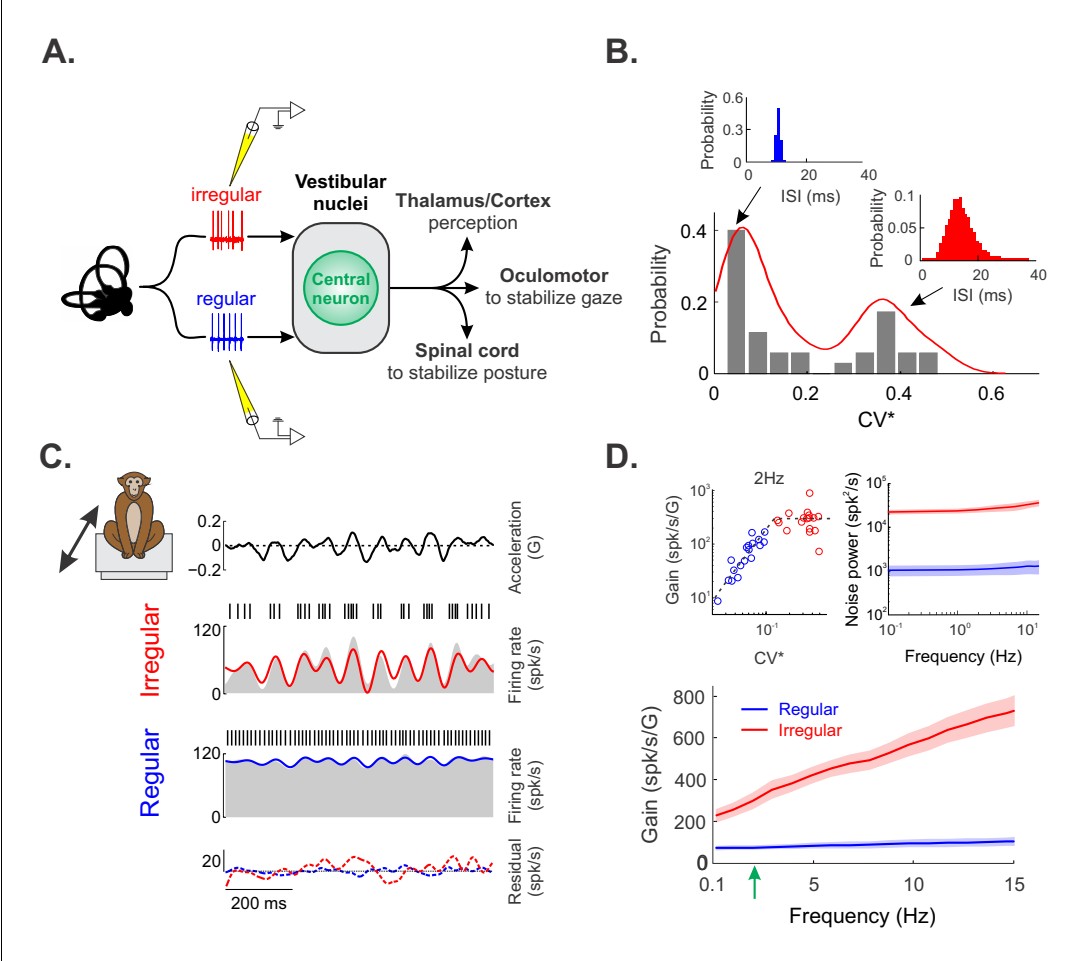

**Figure 1.** Regular and irregular otolith afferent responses to naturalistic translational self-motion stimuli. (A) Schematic showing early vestibular pathways and recording sites. (B) Distribution of resting discharge variability as quantified by CV* for our dataset. As detailed in the Materials and methods, CV* is a normalized coefficient of variation that is used to quantify resting discharge variability independently of differences in firing rate (*Goldberg et al., 1984*). The distribution was clearly bimodal (Hartigan's dip test, p=0.04). The insets show the interspike interval histograms from example regular (left, blue) and irregular (right, red) afferents with CV*=0.06 and 0.29, respectively. (C) Time varying linear head acceleration (top row) with corresponding spiking and firing rate from example irregular (second row from top) and regular (third row from top) afferents. The linear firing rate predictions for the irregular and regular afferents are shown in red and blue, respectively. The time varying residuals (i.e., difference between the actual and predicted responses) are also shown for each afferent (bottom row). (D) Population-averaged gains for regular (blue, N = 18) and irregular (red, N = 17) afferents as a function of stimulus frequency. Top left inset: Gain at 2 Hz (green arrow in bottom panel) as a function of CV*. Top right inset: Population-averaged power spectra of the residual for regular (blue) and irregular (red) afferents. The shaded bands show 1 SEM.

DOI: https://doi.org/10.7554/eLife.45573.002

This raises the question of why irregular afferents have higher residuals than regular afferents. One the one hand, it is possible that nonlinearities in the response of irregular but not regular afferents contribute to the poor fit of standard linear models. On the other hand, irregular afferents could display larger trial-to-trial variability than their regular counterparts, which would also increase the residual. Thus, to determine which of these alternatives is correct, we investigated otolith afferent responses to repeated stimulus presentations. Responses from example irregular and regular afferents are shown in the top and bottom panels of *Figure 2A*, respectively. Notably, responses to repeated stimulus presentations were more similar for the example irregular afferent than for the example regular afferent, as can be seen by the spikes being better aligned across repeated presentations of the stimulus (*Figure 2A*, compare top and bottom raster plots). To test directly for the presence of nonlinearities and to quantify trial-to-trial variability, we computed the response–response (RR) coherence (i.e., the coherence between the responses to repeated stimulus

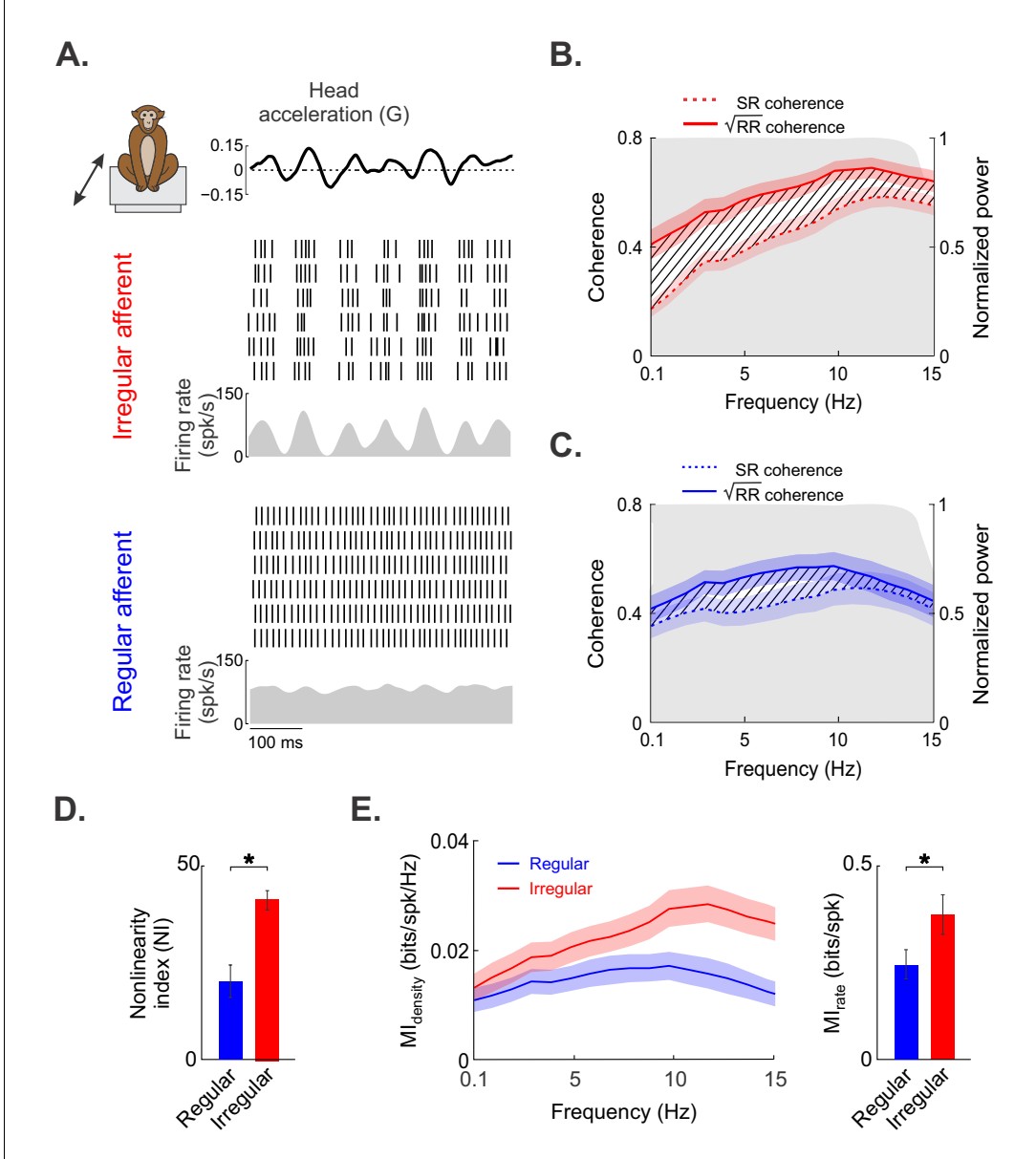

**Figure 2.** Irregular otolith afferents display greater response nonlinearity than their regular counterparts. (**A**) Time-varying stimulus (top) as well as spiking- and firing-rate responses to repeated stimulus presentations, obtained from the same irregular (middle) and regular (bottom) afferents shown in *Figure 1*. (**B**) Population-averaged stimulus–response (SR, dashed red) and square root of the response–response (√RR, solid red) coherence curves for irregular afferents. (**C**) Population-averaged SR (dashed blue) and √RR (solid blue) coherence curves for regular afferents. (**D**) The population-averaged nonlinearity index values for regular (blue) and irregular (red) afferents were significantly different from one another (p=0.0002). (**E**) Population-averaged mutual information density curves for irregular (red) and regular (blue) afferents. Right bar chart: population-averaged mutual information rates for irregular (red) and regular (blue) afferents were significantly different from one another (p=0.04). Note that there is a one-to-one relationship between the mutual information rate density curve and √RR. Moreover, the mutual information rate is obtained by integrating the mutual information rate density over frequency, as detailed in the 'Materials and methods'. '*' indicates statistical significance at the p=0.05 level as determined using a Wilcoxon ranksum test. The shaded color bands around the curves show one SEM.

DOI: https://doi.org/10.7554/eLife.45573.003

presentations) as well the stimulus-response (SR) coherence (i.e., the coherence between the stimulus and the response). The coherence is equal to one if both signals are identical and to 0 if they are uncorrelated. The RR coherence is a measure of the trial-to-trial variability in the response.

Moreover, a difference between the square root of the RR and the SR coherence values indicates the presence of nonlinearities (*Roddey et al., 2000*).

Overall, the square root of the RR coherence was higher for irregular afferents than for regular afferents (*Figure 2B,C*, compare solid red and blue curves), confirming our observation that irregular afferents display lower trial-to-trial variability than their regular counterparts. Therefore, the greater residual observed for irregular afferents cannot be due to increased trial-to-trial variability. We further found greater differences between the square root of the RR and SR coherence curves for irregular afferents (*Figure 2B*, compare dashed and solid red curves) than for their regular counterparts (*Figure 2C*, compare dashed and solid blue curves), indicating greater nonlinearity for the irregular afferents. To quantify this observation, we computed a nonlinearity index (NI, see 'Materials and methods') that is null when both curves are equal and increases with increasing level of nonlinearity. As expected, the population-averaged NI value was significantly higher for irregular afferents (*Figure 2D*; irregular: 40.8 ± 2.2%, regular: 20.0 ± 4.1%; p<0.001, Wilcoxon ranksum test). Therefore, we conclude that the greater residual observed for irregular afferents is due to greater response nonlinearity.

We hypothesized that the greater sensitivity and lower trial-to-trial variability displayed by irregular afferents lead to increased information transmission. To test this prediction, we quantified the mutual information between the neural response and the naturalistic translational self-motion stimulus for irregular and regular afferents (see 'Materials and methods'). Overall, the mutual information rate densities of irregular afferents were consistently higher across frequencies than those of regular afferents (*Figure 2E*, left panel, compare red and blue curves), leading to significantly greater rates of information transmission (*Figure 2E*, right panel; irregular: 0.37 ± 0.05 bits/spk; regular: 0.24 ± 0.04 bits/spk; p=0.04, Wilcoxon ranksum test). Thus, our results show that irregular afferents display stronger nonlinearities but lower trial-to-trial variability in their responses to naturalistic translational self-motion stimuli, which leads to greater information transmission.

## Contributions of spike timing to the encoding of naturalistic translational self-motion stimuli by otolith afferents

Our results above show that irregular afferents displayed lower trial-to-trial variability in their responses to repeated stimulus presentations than their regular counterparts (*Figure 2A*, top panel). This led us to hypothesize that increased information transmission by irregular afferents is due to precise spike timing (i.e., irregular afferents use a temporal code to transmit information). We therefore tested whether the spiking activities in response to different stimulus waveforms were more discriminable from one another at timescales much shorter than those contained in the stimulus waveform for irregular afferents. *Figure 3A* shows the responses of example irregular (red) and regular (blue) afferents to repeated presentations of two different stimulus waveforms (left and right panels). Visual inspection revealed that the responses of the example irregular afferent to both stimulus waveforms were more discriminable from one another compared to those of the example regular afferent (*Figure 3A*, compare red and blue raster plots), which is in part due to their higher sensitivity. Together with the fact that there is lower trial-to-trial variability in the responses of irregular afferents, this implies that the spiking activities that occur in response to different stimulus waveforms are more discriminable from one another at timescales much shorter than those contained in the stimulus waveform for irregular afferents.

In order to quantify this observation, we used metrics to quantify the distance spiking activities between and across stimulus waveforms (see 'Materials and methods'). We first used the Victor–Purpura metric (*Victor and Purpura, 1996*) to quantify the performance of a classifier in determining whether a recorded spiking activity could be correctly predicted as having been elicited by a given stimulus across timescales (see 'Materials and methods'). We predicted that, if information is transmitted via precise spike timing, then discrimination performance should be maximum at timescales significantly shorter than those contained in the stimulus (i.e., <50 ms). Our results confirmed this prediction as discrimination performance was indeed maximum for timescales near 7 ms for irregular afferents (*Figure 3B* (left), *Figure 3—figure supplement 1A*). Qualitatively different results were observed for regular afferents as discrimination performance was not only much lower but also maximum at a much larger timescale of 37 ms, which is closer to those contained in the stimulus (*Figure 3B* (right), *Figure 3—figure supplement 1B*).

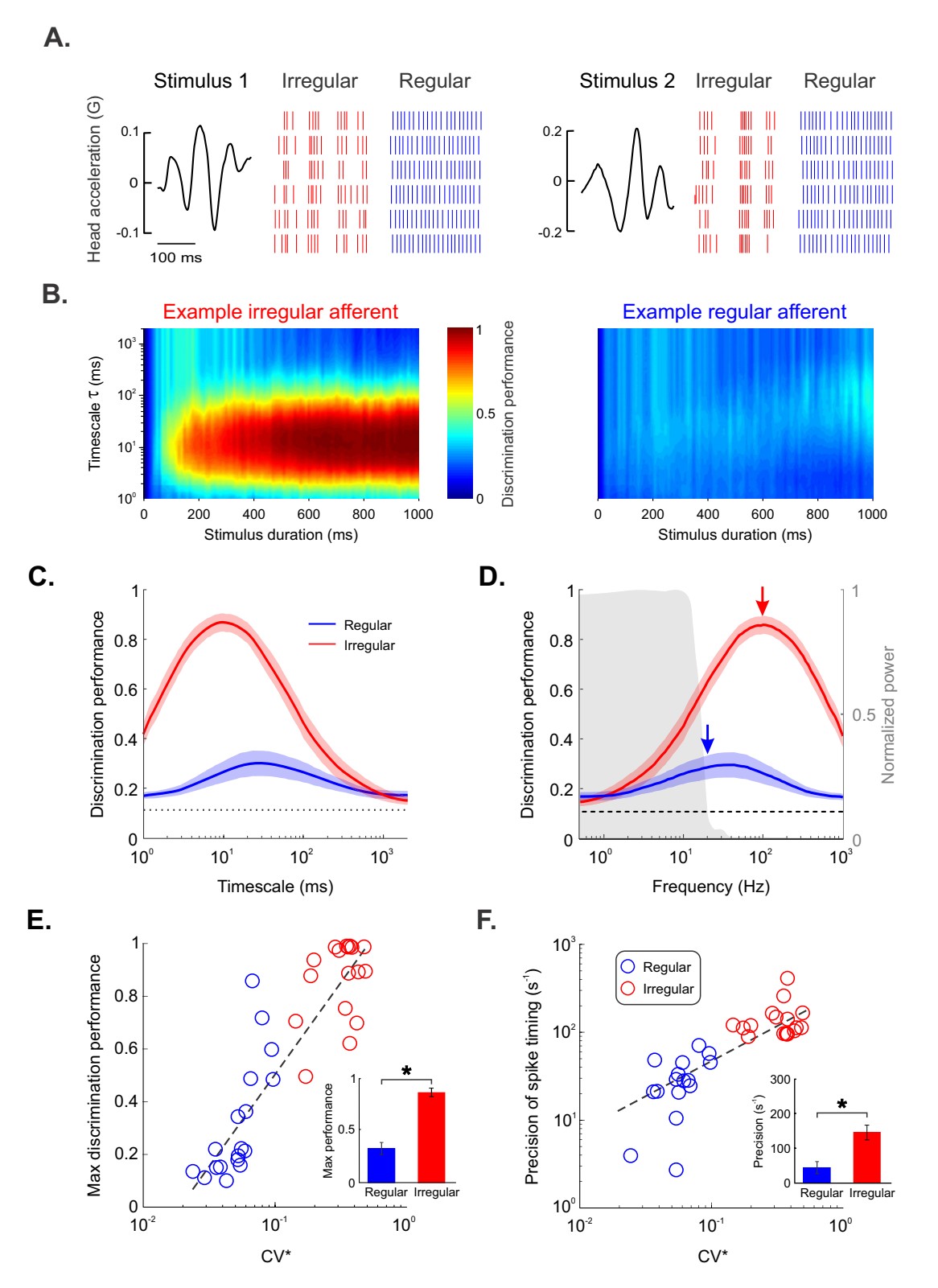

**Figure 3.** Irregular but not regular otolith afferents reliably discriminate between different stimulus waveforms through precise spike timing. (A) Raster plots showing the spiking activities from an example irregular afferent (red) and an example regular afferent (blue) to repeated presentations of two different stimulus waveforms (left and right). (B) Discrimination performance as a function of spike-train duration and timescale $\tau$ (i.e., 1/q, see 'Materials and methods') for the example irregular afferent (left) and regular afferent (right). (C) Population-averaged discrimination performance for

*Figure 3 continued*

irregular (red) and regular (blue) afferents as a function of timescale. Chance performance is also shown (dashed line). The shaded color bands around the curves show one SEM. (D) Population-averaged discrimination performance for irregular (red) and regular (blue) afferents as a function of frequency (i.e., inverse of timescale). Chance performance (dashed line) as well as the stimulus power spectrum (shaded gray) are also shown. The shaded color bands (red and blue) around the curves show one SEM. (E, F) Discrimination performance (E) and precision of spike timing (F) as a function of baseline variability as quantified by CV* for regular (blue) and irregular (red) afferents. Afferents with higher CV* tended to display greater discrimination performance (R = 0.89, p=1.38E-12) as well as higher spike-timing precision (R = 0.71, p=4.69E-06). Insets: Population-averaged discrimination performance (E) and spike-timing precision (F) for regular (blue) and irregular (red) afferents (performance p=3.92E–06, precision p=9.26E–06). '*' indicates statistical significance at the p=0.05 level using a Wilcoxon ranksum test.

DOI: https://doi.org/10.7554/eLife.45573.004

The following figure supplements are available for figure 3:

**Figure supplement 1.** Discrimination performance is strongly linked with resting discharge variability in the otolith afferent population.

DOI: https://doi.org/10.7554/eLife.45573.005

**Figure supplement 2.** Using a different spike train metric (in this case, Van Rossum) does not affect the nature of our results.

DOI: https://doi.org/10.7554/eLife.45573.006

Quantification of our dataset revealed qualitatively similar results for regular and irregular afferents. Indeed, the population-averaged discrimination performance of irregular afferents was considerably higher (three-fold) than that of their regular counterparts across timescales (*Figure 3C*). This difference in performance was not, however, due to differences in firing rates, either during stimulation (regular: 86 ± 5 spk/sec; irregular: 74 ± 7 spk/sec; p=0.16, Wilcoxon ranksum) or at rest (regular: 85 ± 5 spk/sec; irregular: 71 ± 6 spk/sec; p=0.19, Wilcoxon ranksum). Importantly, maximum performance for irregular afferents was observed for timescales of 10 ms (*Figure 3C*) or, equivalently, frequencies of 100 Hz (*Figure 3D*, red arrow). This frequency was five-fold greater than those contained in the naturalistic translational self-motion stimulus (<20 Hz; shaded gray area). By contrast, the discrimination performance for regular afferents reached its maximum value for timescales near 50 ms (*Figure 3C*) or, equivalently, for frequencies at which there is significant stimulus power (~20 Hz; *Figure 3D*, blue arrow). Discrimination performance (*Figure 3E*) and spike-timing precision (i.e., the frequency at which discrimination performance is maximum; *Figure 3F*) were both strongly positively correlated with resting discharge variability, as quantified by CV* (performance: R = 0.89, p<0.001; precision: R = 0.71, p<0.001). Quantitatively similar results were obtained when using a different distance metric (i.e., Van Rossum; *Figure 3—figure supplement 2*). Thus, our findings demonstrate that irregular but not regular otolith afferents use a temporal code to transmit information about naturalistic translational self-motion stimuli.

## Increases in variability and sensitivity lead to greater information transmission and spike-timing precision

To gain understanding as to why regular and irregular otolith afferents exhibited qualitatively different neural coding properties in response to naturalistic translational self-motion stimuli, we built a stochastic mathematical model based on the leaky integrate-and-fire formalism and adjusted parameters so that the spiking output matched the experimental data (*Figure 4A*, see 'Materials and methods'). Overall, we found that, by co-varying the resting discharge variability and sensitivity parameters such that their ratio remains constant (*Figure 4A*, right panel), which is consistent with previous experimental findings (*Jamali et al., 2013*), we were able to reproduce the experimental data from regular and irregular afferents with good accuracy (*Figure 4—figure supplement 1A,B*). We then analyzed the output of our model in the same way as the experimental data above. Specifically, we computed the information transmitted and quantified discrimination performance as well as spike-timing precision (see 'Materials and methods'). Overall, increasing sensitivity and variability in our model increased the information rate (*Figure 4B*), discrimination performance (*Figure 4C*; *Figure 4—figure supplement 1C,D*), and spike-timing precision (*Figure 4D*) to values consistent with those observed experimentally. Thus, our model provides an explanation for our experimental findings. Specifically, our model suggests that the combined effects of the greater sensitivity and resting discharge variability for irregular afferents enable greater information transmission via precise spike timing.

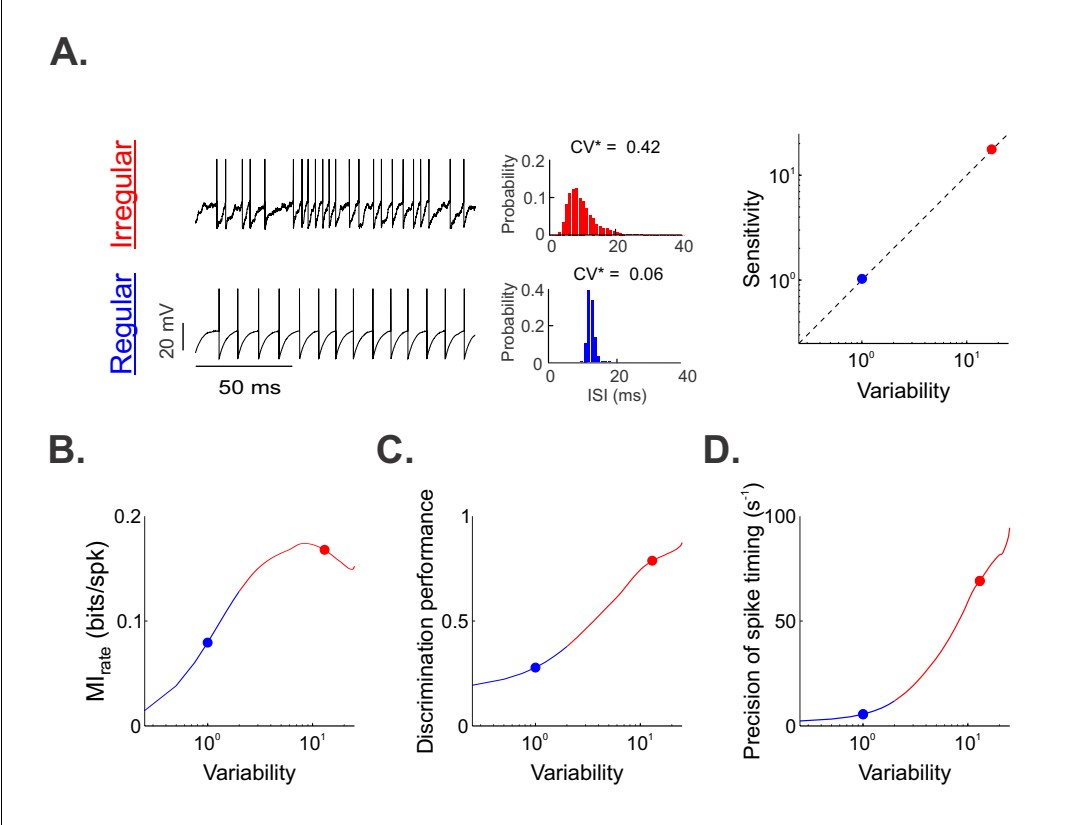

**Figure 4.** A simple mathematical model based on the leaky integrate-and-fire formalism can reproduce our experimental data by co-varying sensitivity and variability. (**A**) ( Left) Example resting (i.e., in the absence of stimulation) spiking activity from our model for parameter values that reproduced data from irregular and regular afferents (CV*=0.06). (Middle) Interspike interval histograms from example irregular (top) and regular (bottom) model afferents with CV*=0.42 and 0.06, respectively. (Right) Variability as a function of sensitivity in our model. Both were co-varied such that their ratio remains unity (dashed line). (**B, C, D**) Mutual information rate (**B**), performance (**C**), and precision (**D**) as a function of variability. The blue and red dots show the values used (**A**) or obtained (**B, C, D**) for the regular and irregular afferent models, respectively. Note that because variability and sensitivity were co-varied in our model, the results obtained in (**B**), (**C**) and (**D**) could be similarly plotted as a function of sensitivity instead of variability.
DOI: https://doi.org/10.7554/eLife.45573.007

The following figure supplement is available for figure 4:

**Figure supplement 1.** Modeling reproduces our experimental data.
DOI: https://doi.org/10.7554/eLife.45573.008

### Irregular otolith afferents display phase-locking to sinusoidal self-motion translational stimuli

As noted above, otolith afferent responses to translational self-motion stimuli have been traditionally characterized using artificial stimuli such as sinusoids. Thus, we next investigated whether our results obtained using naturalistic translational self-motion stimuli also have implications for how otolith afferents encode sinusoidal stimuli. In the auditory system, temporal precision of spiking relative to a sinusoidal stimulus waveform (i.e., a pure tone) is commonly associated with a nonlinear phenomenon termed phase locking. Specifically, phase locking refers to the observation that action potentials will only occur during certain phases of the stimulus cycle (see recent review by *Heil and Peterson, 2017*). Accordingly, we predicted that irregular otolith afferents will preferentially display phase locking to sinusoidal translational self-motion.

We tested these predictions by recording from both irregular (N = 10) and regular (N = 11) afferents during sinusoidal translational self-motion. *Figure 5A* illustrates the responses of example irregular (top panel) and regular (bottom panel) otolith afferents. Overall, consistent with our predictions, the spiking activity of the irregular but not the regular afferent was phase locked to the stimulus (compare top and bottom panels of *Figure 5A*). Indeed, action potentials reliably occurred

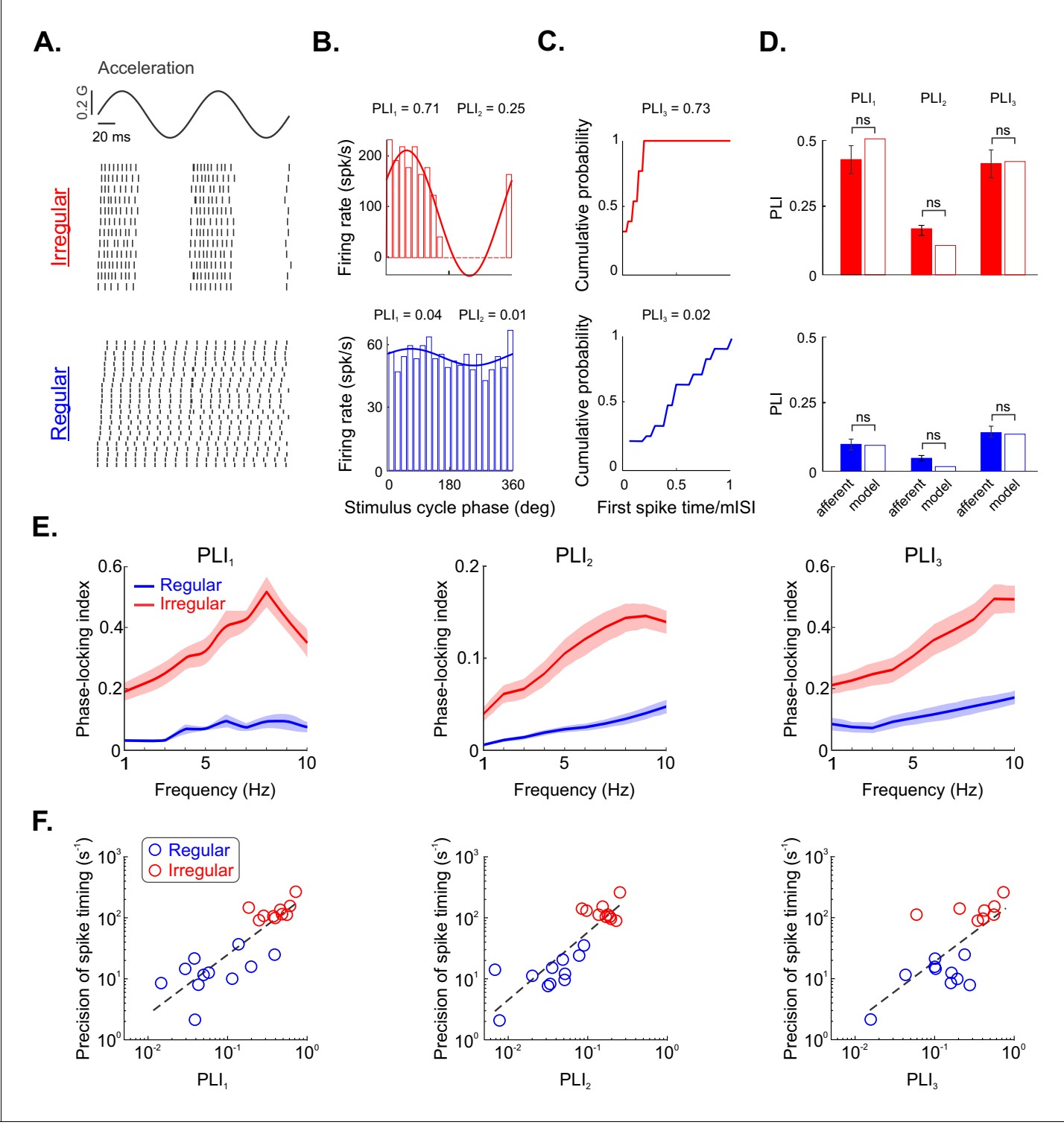

**Figure 5.** Irregular otolith afferents display greater phase locking to sinusoidal stimulation than their regular counterparts. (A) Sinusoidal head acceleration stimulus (top) with raster plots showing the responses of example irregular (middle) and regular (bottom) afferents. (B) Plots of cycle histograms showing firing rate as a function of phase for the same example irregular (top) and regular (bottom) afferents, showing the corresponding values of $PLI_1$ (i.e., vector strength) and $PLI_2$ (i.e., entropy-based). (C) Plots of cumulative probability as a function of first spike time normalized by the mean interspike interval (ISI) for the same example irregular (top) and regular (bottom) afferents with the corresponding values of $PLI_3$. (D) Population-averaged values of $PLI_1$, $PLI_2$, and $PLI_3$ for irregular (top) and regular (bottom) afferents (solid). The hollow bars show the values computed from our model irregular (red) and regular (blue) afferents. Overall, no significant differences were observed (irregular $PLI_1$, p=0.73; irregular $PLI_2$, p=0.55;

*Figure 5 continued on next page*

*Figure 5 continued*

irregular $PLI_3$, p=1; regular $PLI_1$, p=0.83; regular $PLI_2$, p=0.50; regular $PLI_3$, p=1). (E) Phase locking indices $PLI_1$ (left), $PLI_2$ (middle), and $PLI_3$ (right) as a function of stimulus frequency. (F) Spike-timing precision as a function of $PLI_1$ (left), $PLI_2$ (middle), and $PLI_3$ (right) for regular (blue) and irregular (red) afferents. Strong positive correlations were observed in both cases (left: R = 0.84, p=1.88E–06; middle: R = 0.87, p=4.35E–07; right: R = 0.7, p=0.001).
DOI: https://doi.org/10.7554/eLife.45573.009

preferentially during the rising phase of the stimulus for the irregular afferent (*Figure 5A*, top panel). By contrast, for the regular afferent, spiking occurred during all phases of the stimulus cycle (*Figure 5A*, bottom panel). We quantified our results using two commonly used phase-locking index (PLI) measures in the auditory system: $PLI_1$ (i.e., vector strength) and $PLI_2$ (which is based on the entropy of the phase distribution). Moreover, we used a third measure that is based on the latency of the first spike, which has been previously applied to the vestibular system $PLI_3$ (see 'Materials and methods'). Notably, all three measures approach zero when spiking occurs with equal probability at all phases of the stimulus cycle (i.e., there is no phase locking) and approach unity when spiking only occurs at a given stimulus phase (i.e., there is perfect phase locking). Computing PLI values revealed much higher $PLI_1$, $PLI_2$, and $PLI_3$ values for the irregular than for the regular example afferent (*Figure 5B,C*, compare top and bottom panels).

Qualitatively similar results were seen across our dataset as all three *PLI* measures were significantly higher for irregular afferents than for their regular counterparts (*Figure 5D*, solid bars; p<0.001 in all cases, Wilcoxon ranksum tests). Furthermore, we simulated our regular and irregular afferent models and found *PLI* values that were comparable to those obtained experimentally (*Figure 5D*, compare solid and hollow bars; irregular — p=0.73, p=0.55, and p=1 for $PLI_1$, $PLI_2$, and $PLI_3$, respectively; regular — p=0.83, p=0.50, and p=1 for $PLI_1$, $PLI_2$, and $PLI_3$, respectively; Wilcoxon ranksum tests). Further, we quantified phase locking in our experimental data for different stimulation frequencies. Overall, irregular afferents displayed increased tendency to phase lock with increasing stimulation frequency, and this tendency was consistently larger than that of their regular counterparts (*Figure 5E*). Finally, to test our prediction that increased spike-timing precision during naturalistic translational self-motion leads to greater phase locking during sinusoidal translational self-motion, we plotted spike-timing precision as a function of phase locking as quantified by *PLI* measures (*Figure 5F*). Spike-timing precision was strongly positively correlated with each measure of phase locking (R = 0.84, 0.87, and 0.7 using $PLI_1$, $PLI_2$, and $PLI_3$, respectively; p≤0.001 in all cases). Thus, taken together, our results firmly establish that information transmission through the precise spike timing observed in irregular otolith afferents is tightly linked to their propensity to display phase locking in response to sinusoidal stimulation. We further consider the implications of this finding in the 'Discussion'.

## Regular otolith afferents better encode differences in static head orientation relative to gravity than their irregular counterparts

To summarize thus far, we have shown that irregular afferents transmit more information overall about naturalistic translational self-motion stimuli than do their regular counterparts, in part through precise spike timing. Our modeling predicted that this was due to the increased resting discharge variability and sensitivity of irregular afferents, which causes phase locking in response to sinusoidal stimulation. So why have regular afferents at all? We note that, thus far, our analysis only considered dynamic translational self-motion stimuli and not the gravito-inertial forces experienced when the head is stationary at different orientations relative to gravity.

Thus, we next recorded the activities of regular and irregular afferents with the animal's head initially upright relative to gravity (i.e., the reference orientation) and then with animal's head positioned stationary at orientations from 3° to 15°, thereby causing changes in the net acceleration that is sensed by otolith afferents (*Figure 6A*). We then compared the resulting firing rate distributions obtained under the different conditions (see 'Materials and methods'). *Figure 6A* shows the firing rate distributions for example regular (blue) and irregular (red) afferents when the animal's head is positioned at 3° (top panel) versus 15° (bottom panel) relative to earth vertical (black). To quantify discriminability between the firing rate distributions obtained when the animal's head is positioned stationary at different orientations and those obtained for earth vertical, we used

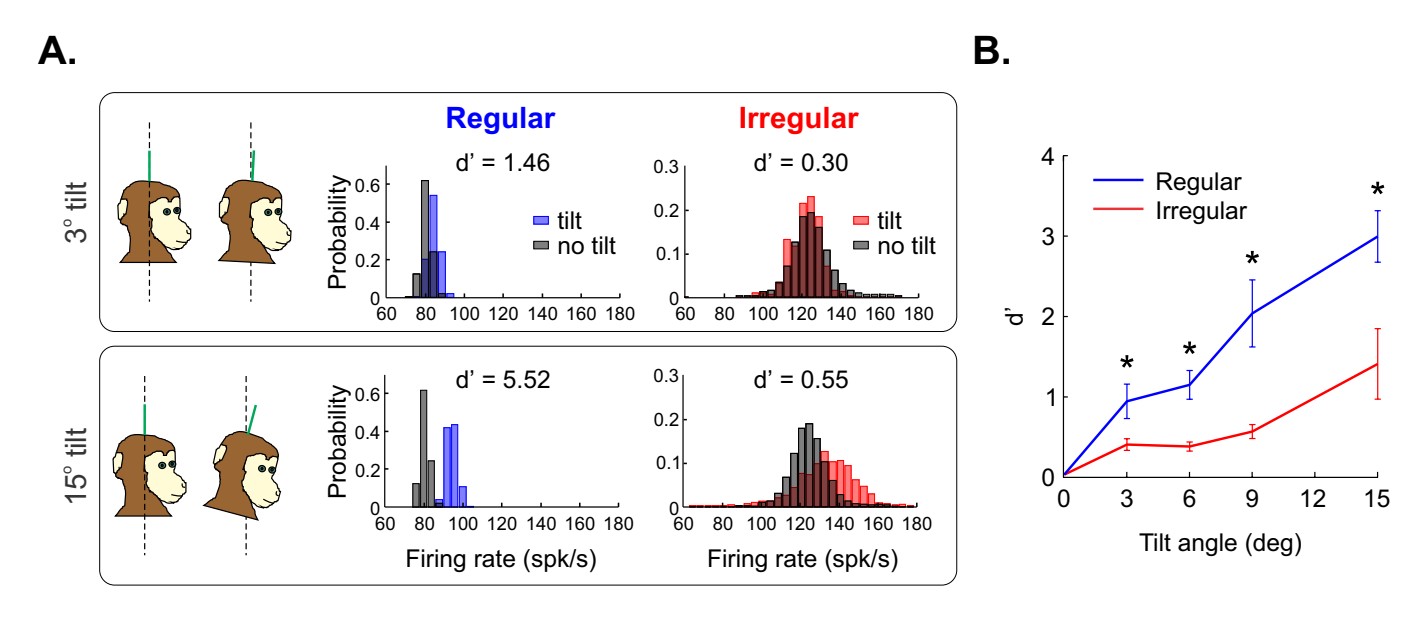

**Figure 6.** Regular otolith afferents better discriminate between different head orientations relative to gravity than do their irregular counterparts. (**A**) (Top left) schematic showing the head positioned vertically (i.e., 0°, left) and at a 3° angle relative to gravity (right). (Top center) Firing rate distributions from an example regular afferent when the head is positioned at 0° (black) and at 3° (blue). (Top right) Firing rate distributions from an example irregular afferent when the head is positioned at 0° (black) and at 3° (red). Also shown are the values of discriminability d'. (Bottom) Same as top row but with a head position at a 15° angle relative to gravity. Firing rates were obtained by convolving the spike trains with a Gaussian spike-density function with standard deviation 10 ms. Firing-rate distributions were obtained using a binwidth of 4 spk/sec (as detailed in the 'Materials and methods'). (**B**) Population-averaged discriminability index d' values for regular (blue) and irregular (red) otolith afferents as a function of head orientation angle. '*' indicates statistical significance at the p=0.05 level using a Wilcoxon ranksum test.

DOI: https://doi.org/10.7554/eLife.45573.010

signal detection theory (*Green and Swets, 1966*; see 'Materials and methods'). We found that the example irregular afferent showed poor discriminability at 3°, which only slightly improved at 15°, as quantified by d' values of 0.3 versus 0.55, respectively (compare top right and bottom right panels of *Figure 6A*). By contrast, the example regular afferent's firing rate distribution measured when the head was positioned at 3° was already discriminable from that measured under reference orientation as quantified by a high d' value of 1.46 (*Figure 6A*, top center panel). Discriminability further increased when considering a 15° angle as both distributions displayed minimal overlap as quantified by a d' value of 5.52 (*Figure 6A*, bottom center panel). This finding was consistent across our population of otolith afferents as discriminability was significantly higher for regular afferents than for irregular afferents across all angles (Wilcoxon Ranksum test, p<0.03 in all cases; *Figure 6B*). These results show that the activities of regular otolith afferents provide a greater ability to discriminate between different static head orientations than is provided by their irregular counterparts.

## Discussion

Here we show that heterogeneities in the otolith afferent population give rise to different coding strategies to represent the gravito-inertial forces experienced during dynamic naturalistic translational self-motion and static head orientation relative to gravity. Analysis of responses to naturalistic translational self-motion stimuli revealed that irregular afferents displayed stronger nonlinearities, and transmitted more information than their regular counterparts for frequencies above 0.1 Hz, via both changes in firing rate and precise spike timing. By contrast, regular afferents transmitted less information primarily through changes in firing rate. Mathematical modeling reproduced these experimental findings, and further predicted that irregular afferents will display stronger phase locking in response to sinusoidal stimulation than their regular counterparts. We validated this prediction experimentally, thereby establishing a tight link between the mechanisms underlying precise spike

timing in irregular otolith afferents and their propensity to phase lock in response to sinusoidal stimulation. Finally, using signal detection theory, we found that regular afferent spiking activities obtained by statically positioning the head at different orientations relative to gravity were more discriminable from one another than for irregular afferents. Taken together, our findings establish that irregular and regular otolith afferents differentially encode naturalistic translational head motion and static orientation relative to gravity. Specifically, irregular otolith afferents with high sensitivity and resting discharge variability preferentially encode translational head motion over the 0.1–15 Hz frequency range, whereas regular otolith afferents with lower sensitivity and resting discharge variability instead preferentially encode static head orientation relative to gravity.

## Role of resting discharge variability in determining coding in the otolith system

Our results show that irregular afferents, which have greater resting discharge variability than their regular counterparts, actually show lower trial-to-trial variability during naturalistic translational self-motion stimulation. We hypothesize that this is due to the observed strong nonlinearities in their responses, which are due at least in part to higher sensitivity. As such, our results show that the coding properties of otolith afferents during naturalistic translational self-motion stimulation are qualitatively different than those observed previously, which solely considered changes in firing rate (*Jamali et al., 2013*; *Yu et al., 2015*). Thus, while theory predicts that trial-to-trial variability during stimulation will increase with increasing resting discharge variability for linear systems (*Chacron et al., 2003*; *Risken, 1996*), this is not valid when strong response nonlinearities are present, as is the case during naturalistic translational self-motion. Nevertheless, it is important to note that the relatively lower intensity stimuli resulting from statically positioning the head at different orientations relative to gravity do not elicit strong response nonlinearities from either regular or irregular afferents (*Fernandez et al., 1972*). As a result, the trial-to-trial variability during static stimulation will be proportional to the resting discharge variability, and thus lower for regular afferents relative to their irregular counterparts. Our results show that this lower resting discharge variability offsets the detrimental effects of lower sensitivity, thereby giving rise to greater stimulus discrimination.

## Differential encoding of gravito-inertial forces by irregular and regular otolith afferents: implications for perception

Our results show that regular and irregular otolith afferents use different coding strategies to provide estimates of both dynamic head motion and static orientation relative to gravity. Overall, the mutual information rate densities of irregular afferents were consistently higher across almost the entire physiologically relevant frequency range (0.1–15 Hz) than those of regular afferents. However, as we were only able to reliably estimate information for frequencies ≥0.1 Hz (i.e., the inverse of the stimulus duration of 10 s), how regular and irregular otolith afferents respond to naturalistic translational self-motion for frequencies <0.1 Hz remains unknown. Nonetheless, on the basis of our results obtained using static head orientations (i.e., 0 Hz), we predict that the information transmitted by regular afferents will become greater than that transmitted by irregular afferents as the frequency approaches zero (*Figure 2E*). This implies that regular afferents outperform their irregular counterparts at encoding low-frequency (i.e., <0.1 Hz) head acceleration signals. Further experiments are needed to test this hypothesis. It should be noted that otolith afferents cannot distinguish the forces that result from translational self-motion from those resulting from changes in head orientation, as detailed below. Therefore, we predict that our results will apply equally to stimuli resulting from either condition. Specifically, we predict that irregular otolith afferents will outperform their regular counterparts at encoding dynamic changes in head orientation, provided that their temporal frequency content is high enough (i.e., >0.1 Hz). Moreover, we predict that regular otolith afferents will outperform their irregular counterparts at encoding low frequency (i.e., <0.1 Hz) translational self-motion.

Behavioral studies have shown that humans and monkeys can distinguish translational self-motion direction when the acceleration exceeds ~1 cm/s$^2$ (s *MacNeilage et al., 2010*, *Soyka et al., 2011*, *Valko et al., 2012* and *Bermúdez Rey et al., 2016*; whereas those on monkeys include *Gu et al., 2007* and *Yu et al., 2015*). However, neural detection thresholds for single otolith afferents are substantially higher (~10 cm/s$^2$) (*Jamali et al., 2013*; *Yu et al., 2015*), indicating that pooling the

activities of multiple otolith afferents is required to give rise to perception of translational self-motion. By contrast, when considering differences in static head orientation relative to gravity, our results here have shown that single regular otolith afferents display discriminability values that are similar to those reported in psychophysical studies (~2°) (*Clemens et al., 2011*; *Valko et al., 2012*; *Karmali et al., 2014*; *Janssen et al., 2011*; *Dahlem et al., 2016*; *Tarnutzer et al., 2013*). This suggests that little additional pooling of afferent activities is actually required. We speculate that this exceptional sensitivity to static differences in spatial orientation serves an essential role in the maintenance of posture during everyday life.

An important problem for the vestibular system is how to distinguish the gravito-inertial forces that result from translational self-motion from those that result from changes in head orientation (also commonly referred to as tilts). Einstein's equivalence principle posits that the forces resulting from both conditions are physically indistinguishable from one another (*Angelaki and Cullen, 2008*; *Guedry, 1974*; *Young, 2011*). Thus, otolith afferents, which sense linear acceleration, cannot be used alone to distinguish between both conditions (see *Goldberg, 2012*). It is important to note, however, that semicircular canal afferents will be stimulated by the rotations associated with changing head orientation relative to gravity but not by translational self-motion. Accordingly, the brain can theoretically distinguish between these two movement conditions by integrating multi-sensory inputs from both otolith and canal afferents (*Guedry, 1974*; *Young, 2011*; *Mayne, 1974*; *Angelaki and Yakusheva, 2009*; *Merfeld et al., 1999*).

To date, the neuronal mechanisms underlying the distinction between the gravito-inertial forces resulting from tilts versus those resulting from translations have been investigated with well-established methods that use stimuli with identical linear accelerations (*Laurens et al., 2013*; *Laurens et al., 2011*; *Angelaki et al., 2004*; *Yakusheva et al., 2007*; *Yakusheva et al., 2008*). Notably, Purkinje cells in the caudal vermis integrate otolith and semicircular canal inputs (reviewed in *Angelaki and Cullen, 2008*), such that one subset preferentially encodes translations (*Yakusheva et al., 2007*; *Yakusheva et al., 2008*) while another preferentially encodes tilts (*Laurens et al., 2013*). Parietoinsular vestibular cortex neurons can likewise discriminate tilts from translations (*Liu et al., 2011*). Interestingly, translational self-motion at very low frequencies (i.e., <~0.1 Hz) is incorrectly interpreted as a change in head orientation (*Glasauer and Merfeld, 1997*; *Kaptein and Van Gisbergen, 2006*; *Seidman et al., 1998*; *Merfeld et al., 2005b*; *Merfeld et al., 2005a*), thereby generating the 'somatogravic' illusion, a cause of disorientation that can be extremely dangerous for pilots. The firing rate modulation of tilt-sensitive cerebellar Purkinje cells appears to provide a neural substrate for this perceptual effect (*Laurens et al., 2013*). Further studies will be needed to investigate whether, and if so how, changes in firing rate and precise spike-timing information from regular and irregular otolith afferents are differentially decoded by downstream pathways to give rise to other attributes of perception and/or behavior. For example, approaches dissociating the contributions of regular and irregular afferents (e.g., silencing of irregular afferents via anodal currents; *Minor and Goldberg, 1991*) should provide new insights.

## Mechanisms underlying spike-timing precision and phase locking in otolith afferents

Our present results further show that the precise spike timing that is preferentially displayed by irregular otolith afferents in responses to naturalistic translational self-motion is strongly correlated with a tendency to phase lock in response to sinusoidal stimulation. In this context, it is important to note that prior studies investigating otolith afferent responses have predominately focused on relatively low frequencies (of <5 Hz) (*Fernández and Goldberg, 1976b*; *Angelaki and Dickman, 2000*; *Yu et al., 2012*; but see *Jamali et al., 2013*), whereas natural translational self-motion stimuli contain frequencies up to 20 Hz (*Carriot et al., 2014*; *Carriot et al., 2017*). Here, we found that irregular otolith afferents show significant phase locking that increases as a function of stimulation frequency within the physiologically relevant range. Interestingly, auditory stimuli can also induce phase locking in otolith afferents, a property that may relate to the otoliths' evolutionary origin as hearing organs (reviewed in *Straka et al., 2016*). Specifically, phase locking has been reported in irregular otolith afferents for sound and vibration stimuli (reviewed in *Curthoys and Grant, 2015*) whose frequency content is much higher (100–3,000 Hz) than that found in natural vestibular stimuli (0–20 Hz; *Carriot et al., 2014*; *Carriot et al., 2017*). It has been proposed that such high-frequency stimuli

cause fluid pressure waves in vestibular end organs, which in turn activate vestibular hair cells (*Eatock and Songer, 2011*; *Curthoys and Grant, 2015*).

The strong correlation between spike-timing precision in response to naturalistic stimulation and phase locking to sinusoidal stimulation seen for otolith afferents suggests that these two processes share common underlying mechanisms. Neurons in the auditory system also display precise spike timing, which is commonly associated with phase locking (see recent review by *Heil and Peterson, 2017*). Such phase locking is mediated by low-threshold voltage-gated potassium channels that promote hyperpolarization after spiking (*Oertel et al., 2000*; *Day et al., 2008*; *Kuznetsova et al., 2008*; *Rothman and Manis, 2003*). Interestingly, such channels are also present in vestibular afferents (*Goldberg et al., 1984*; *Kalluri et al., 2010*; *Highstein and Politoff, 1978*; *Iwasaki et al., 2008*), suggesting that the same mechanism mediates both phase locking and spike-timing precision in the auditory and vestibular systems. Further, morphological differences between regular and irregular afferents could also contribute to the observed differences in phase locking and spike-timing precision. Notably, regular afferents integrate information from multiple hair cells located far away from the spike initiation zone (*Goldberg, 1991*), whereas irregular afferents integrate synaptic input from relatively few hair cells and have their spike initiation zone located close by (*Highstein and Politoff, 1978*). The latter configuration probably minimizes spatial and temporal integration and is therefore predicted to lead to more precise action potential firing, consistent with our present findings.

## The vestibular system uses different strategies to encode rotational and translational self-motion stimuli

Another important consequence of our results is that they firmly establish significant differences in how the otolith and canal systems encode naturalistic translational and rotational self-motion, respectively. Specifically, canal afferents also display heterogeneities in resting discharge and can also be classified as either regular or irregular (see *Goldberg, 2000* for review). However, the greater sensitivity of irregular canal afferents is not sufficient to compensate effectively for their higher resting discharge variability (*Sadeghi et al., 2007*). Accordingly, irregular canal afferents display lower detection thresholds than their regular counterparts over the entire behaviorally relevant frequency range (*Jamali et al., 2016*; *Massot et al., 2011*; *Sadeghi et al., 2007*), while the detection thresholds of regular and irregular otolith afferents are similar over this range of frequencies (*Jamali et al., 2013*).

Comparison between the results presented here, which are focused on otolith afferents, and those of prior studies, which are focused on canal afferents, further reveals fundamental differences in information-coding strategies. Specifically, regular canal afferents preferentially encode rotational self-motion through changes in firing rate, whereas irregular canal afferents preferentially encode these through precise spike timing (*Jamali et al., 2016*). By contrast, our present results show that irregular otolith afferents transmit more information about high frequency (>0.1 Hz) translational self-motion than their regular counterparts through both changes in firing rate and precise spike timing, whereas regular afferents transmit more information about static head orientation relative to gravity through changes in firing rate. In this context, it is important to note that while otolith afferents display sustained responses to static forces (such as gravity) (*Angelaki and Dickman, 2000*; *Fernández and Goldberg, 1976a*; *Fernandez et al., 1972*; *Purcell et al., 2003*), this is not the case for canal afferents as their sustained responses to rotations at constant angular velocities are minimal (*Fernandez and Goldberg, 1971*). Taken together, our present results thus highlight the need for further work to understand how the distinctive coding strategies used by peripheral otolith versus canal afferents are decoded by downstream neurons in order to process these different attributes of head motion (e.g., linear motion vs. head orientation).

## Materials and methods

All experimental protocols were approved by the McGill University Animal Care Committee (#2001–4096) and were in compliance with the guidelines of the Canadian Council on Animal Care.

## Surgical preparation

Two male macaque monkeys (*Macaca fascicularis*), aged 6 and 8 years old, were prepared for chronic extracellular recording under aseptic conditions. The surgical preparation was similar to that previously described (*Dale and Cullen, 2013*). Briefly, under isoflurane anesthesia (0.8–1.5%), a stainless steel head post was secured to the animal's skull with stainless steel screws and dental acrylic, allowing complete immobilization of the head during the experiments. The implant also held in place a recording chamber oriented stereotaxically towards the vestibular nerve where it emerges from the internal auditory meatus. Finally, an 18 mm diameter eye coil (three loops of Teflon-coated stainless steel wire) was implanted in the right eye behind the conjunctiva. After the surgery, buprenorphine was administered as analgesic (0.01 mg/kg IM, every 12 hr for 2–5 days) and cefazolin (25 mg/kg IM) was injected twice daily for 10 days. Animals were given at least 2 weeks to recuperate from the surgery before any experiments began and were housed in pairs on a 12 hr light/dark cycle.

## Data acquisition

During experiments, the head-fixed monkey was seated in a primate chair mounted on top of a linear actuator in a dimly lit room. Thus, all vestibular stimulation was passively generated by the linear actuator rather than actively by the animal. The vestibular nerve was approached through the floccular lobe of the cerebellum, as identified by its eye-movement-related activity (*Lisberger and Pavelko, 1986*; *Cullen and Minor, 2002*); entry to the nerve was preceded by a silence, indicating that the electrode had left the cerebellum. Extracellular single-unit activity of otolith afferents was recorded using glass microelectrodes (24–27 MΩ) as previously described (*Jamali et al., 2009*). Head acceleration was measured by a 3-D linear accelerometer (ADXL330Z, Analog Devices, Inc, Norwood, MA) firmly secured to the animal's head post. During experimental sessions, unit activity, horizontal and vertical eye positions, and head acceleration signals were recorded on digital audiotape for later playback. During playback, action potentials from extracellular recordings were discriminated using a windowing circuit (BAK Electronics, Mount Airy, MD). Eye position and head acceleration signals were low-pass filtered at 250 Hz (eight-pole Bessel filter) and sampled at 1 kHz. Data were imported into Matlab (The MathWorks, Natick MA) for analysis using custom-written algorithms (*Jamali et al., 2019*).

## Experimental design

The otolith afferents included in the present study were characterized on the basis of their response to sinusoidal translational head movements (5 Hz, 0.2 G; G = 9.8 m/s$^2$) applied along the fore-aft (90°) and/or lateral (0°) axes, and the absence of modulation during yaw rotations. Owing to limitations of our experimental setup, afferents that were predominantly sensitive to stimulation along the vertical axis were not included in our dataset.

Once an otolith afferent fiber was isolated, we first determined its preferred direction (PD), which is the axis along which the neuron is maximally responsive (*Jamali et al., 2013*). First, we applied sinusoidal translation (5 Hz, 0.2 G) along the fore-aft and lateral axes in the horizontal plane, and the sensitivity of the unit was computed in both directions using Spike2 software (CED, Cambridge, UK) and custom-written MATLAB algorithms (*Jamali et al., 2019*). We then used these measurements to estimate the tuning curve (i.e. the sensitivity as a function of direction) using a cosine fit (*Angelaki and Dickman, 2000*; *Purcell et al., 2003*; *Fernández and Goldberg, 1976a*). The preferred direction was then taken as the orientation for which neuronal sensitivity was maximal.

Each individual afferent was stimulated along its PD using three types of linear self-motion stimuli.

(i) Sinusoidal translations at 10 different frequencies (1–10 Hz) with peak acceleration of 0.2 G, which were used to investigate phase-locking behavior in otolith afferents.

(ii) Different head orientations relative to gravity obtained by statically re-aligning the whole body at 3°, 6°, 9°, and 15° pitch angles in the direction that increased the afferent's activity (i.e., nose up or nose down). After a waiting period following the applied change in pitch angle (*Fernández and Goldberg, 1976a*), we recorded each afferent's activity for a minimum of 10 s while the head was statically positioned in the new orientation.

(iii) Naturalistic translational self-motion stimuli that consisted of low-pass filtered Gaussian white noise (20 Hz cutoff) with zero mean and standard deviation of 0.1 G. The same 10 s realization of the stimulus was repeated 10 times on average in order to assess trial-to-trial variability.

We note that, due to the finite duration of the stimulus, we could not reliably estimate information for frequencies below 0.1 Hz (i.e., the inverse of the stimulus duration). Furthermore, estimating the mutual information at lower frequencies (e.g., 0.01 Hz) would require holding the afferent for durations of 1000 s (i.e., >16 min), which is not currently feasible given the vigorous nature of naturalistic self-motion stimuli.

## Analysis of neuronal discharges

### Resting discharge

The regularity of resting discharge (i.e., in the absence of stimulation) was determined by means of a normalized coefficient of variation (CV*, after *Goldberg et al., 1984*) of the interspike intervals (ISIs) recorded during spontaneous activity. Afferents with low values of CV* were classified as regular, whereas those high values of CV* (see *Figure 1*) were classified as irregular as in previous studies (*Goldberg et al., 1990*; *Jamali et al., 2009*; *Yu et al., 2012*).

### Information transmission through changes in firing rate

Neural firing rates $fr(t)$ were estimated by convolving the spike trains with a Gaussian spike density function (standard deviation of 10 ms) as previously described (*Roy and Cullen, 2001*). To estimate the response gain, the time varying firing rate $fr(t)$ and the stimulus $S(t)$ (i.e., linear acceleration) were both sampled at 1000 Hz. A transfer function $H(f)$ was computed using $H(f)=P_{Sfr}(f)/P_{SS}(f)$ and the response gain was then computed as the magnitude of the transfer function $G(f)=|H(f)|$, where $P_{Sfr}(f)$ is the cross-spectrum between the stimulus $S(t)$ and the firing rate $fr(t)$, and $P_{SS}(f)$ is the power spectrum of the stimulus $S(t)$. All spectral quantities (i.e. power-spectra, cross-spectra) were estimated using multitaper estimation techniques with eight Slepian functions (*Jarvis and Mitra, 2001*) as previously described (*Sadeghi et al., 2007*). We obtained a linear estimate of the firing rates as described previously (*Massot et al., 2012*). This was done by convolving the transfer function $H(t)$ described above with the stimulus $S(t)$ and adding the baseline firing rate to this in order to form the linear prediction. The residual $N(t)$ was then computed as the difference between the actual firing rate and its linear estimate.

### Response nonlinearity and phase locking

We quantified correlations between the spike train and the stimulus. To do so, the spike train of each afferent in response to the noise stimuli was converted into a binary sequence $R(t)$ with a bin width of 1 ms. The value of each bin was set to one if it contained an action potential and 0 otherwise. The stimulus-response (SR) coherence $C_{SR}(f)$ between the binary sequence $R(t)$ and the stimulus $S(t)$ was computed as in previous studies (*Roddey et al., 2000*):

$$C_{SR}(f) = \frac{|P_{SR}(f)|^2}{P_{SS}(f)P_{RR}(f)} \tag{1}$$

where $P_{SR}(f)$ is the cross-spectrum between $S(t)$ and $R(t)$, and $P_{SS}(f)$ and $P_{RR}(f)$ are the power spectra of $S(t)$ and $R(t)$. The SR coherence varies between 0 and 1 and quantifies the extent to which $S(t)$ and $R(t)$ are linearly correlated for a given frequency $f$. To explore the presence of nonlinearity in the responses of otolith afferents, we quantified the coherence between neuronal activities in response to repetitions of the same stimulus. The response–response coherence (RR coherence) between sequences of action potentials was computed by:

$$C_{RR}(f) = \frac{|<P_{R_iR_j}(f)>_{i \neq j}|^2}{(<P_{R_iR_i}(f)>_i)^2} \tag{2}$$

where $P_{R_iR_j}(f)$ is the cross-spectrum between binary sequence $R_i(t)$ and $R_j(t)$, and $P_{R_iR_i}(f)$ is the power spectra of $R_i(t)$, respectively. Note that the average $<\ldots>_{i,j}$ is over all possible combinations of $i$ and $j$ where $j < i$, while $<\ldots>_i$ is the average over index $i$. $C_{RR}(f)$ is also a number between 0 and 1 and signifies the degree to which the responses to repeated presentations of the same stimulus

are correlated at frequency f (**Roddey et al., 2000**). For k repetitions of the stimuli, the equation above becomes:

$$C_{RR}(f) = \frac{\left| \frac{2}{k(k-1)} \sum_{i=2}^{k} \sum_{j=1}^{i-1} P_{R_i R_j}(f) \right|^2}{P_{RR}(f)^2} \tag{3}$$

Since in general $\sqrt{C_{RR}(f)} \geq C_{SR}(f)$, a linear model is optimal if the *SR* coherence equals the square root of the *RR* coherence. A significant difference between these two quantities indicates that a nonlinear model is necessary to explain the relationship between the stimulus *S(t)* and the response *R(t)* for a given frequency f (**Roddey et al., 2000**). To quantify the difference between the *SR* and *RR* coherence estimates, we computed a non-linearity index (NI) as previously described (**Chacron, 2006**):

$$NI = 100 \times \left( 1 - \frac{\int_0^{100} C_{SR}(f)\, df}{\int_0^{100} \sqrt{C_{RR}(f)}\, df} \right) \tag{4}$$

A perfectly linear response results in an NI of zero, whereas with increasing non-linearity, NI approaches 100%.

We calculated the mutual information rate density between the stimulus *S(t)* and R(t) using (**Rieke et al., 1996**; **Sadeghi et al., 2007**; **Theunissen et al., 1996**):

$$MI_{density}(f) = -\log_2 \left[ 1 - \sqrt{C_{RR}(f)} \right] / FR \tag{5}$$

where *FR* is the mean firing rate during stimulation. This normalization accounts for the fact that the mutual information rate density increases with firing rate (**Borst and Haag, 2001**). The mutual information rate was obtained by integrating (i.e., computing the area under the curve) the mutual information rate density over the frequency range 0–15 Hz.

To measure the phase-locking behavior of otolith afferents, we computed the phase-locking index (PLI) at each frequency of sinusoidal translation using methods previously used in the vestibular literature. As our first method, we calculated vector strength or synchronization index according to the following equation (**Goldberg and Brown, 1969**):

$$PLI_1 = \frac{\sqrt{\left[ \sum_{i=1}^{n} \cos(\theta_i) \right]^2 + \left[ \sum_{i=1}^{n} \sin(\theta_i) \right]^2}}{n} \tag{6}$$

where $\theta_i$ is the phase angle of spike *i* relative to the modulation cycle of the stimulus, and n is the total number of action potentials in the analysis window. $PLI_1$ can vary between 0 and 1, with one indicating a perfect entrainment between the neuronal response and the modulation phase and 0 signifying no correlation.

We further used a second measure of phase locking that was based on the entropy of the cycle histogram (**Kajikawa and Hackett, 2005**; **Schneider et al., 2011**) which, unlike measures of vector strength (**Mardia and Jupp, 2000**), can quantify the degree of phase locking even in multi-peaked phase histograms, as in our case. The phase-locking index was quantified as:

$$PLI_2 = 1 - E_0/E_{max} \tag{7}$$

$$E_0 = -<P(\varphi)\log_2 P(\varphi)>$$

$$E_{max} = \log_2 N_{bin}$$

where $P(\varphi)$ is the probability of firing a spike as a function of stimulus phase, $E_0$ is the entropy of the phase probability distribution, and $E_{max}$ is the maximum entropy that corresponds to a uniform distribution. Note that $PLI_2$ can vary between 0 and 1. When comparing *PLI* values to spike-timing precision, we used values computed in response to 10 Hz sinusoidal stimulation. We also used a third

method based on the latency of the first spike (*Ramachandran and Lisberger, 2006*). Rasters of action potentials corresponding to each cycle of the stimulus were ordered according to the time at which the first spike was elicited. Then by plotting the time of the first spike as a function of the spike train number in the ordered rasters, the phase-locking index was computed as:

$$PLI_3 = 1 - \frac{\rho N}{\mu_{ISI}}$$

(8)

where ρ is the slope of the relationship between the time of the first spike and the binary sequence number, N is the number of spike trains in the raster, and $\mu_{ISI}$ PLI3 represents the mean interspike interval during resting discharge. In essence, this measure determines what fraction of the average ISI ($\mu_{ISI}$ PLI3) is covered by the range of first spike times ($\rho N$ PLI3) from the responses to each cycle of the stimulus. If the times of the first spikes remain relatively constant (negligible ρ), such that their range only extends up to a small fraction of the average ISI, then $PLI_3$ approaches 1, indicating that the neuron exhibits phase-locking. On the other hand, if the range of first spike times expands across the rasters and reaches the average ISI, the value of $PLI_3$ is near zero, which means that the spikes are not phase locked to the stimuli.

## Firing rate responses in different spatial orientations

For each afferent, we determined the degree to which an ideal observer can distinguish an arbitrary non-zero acceleration $\hat{H}^A$ from null (i.e, when the animal's head was upright relative to gravity) using their corresponding empirical firing-rate distributions obtained for different head orientations. Specifically, we removed the first and the last 2 seconds of neural activity recorded while the head was statically positioned in a given orientation and used the time-dependent firing rate *fr(t)* to compute its probability density using a firing-rate bin-width of 4 spk/sec. We then computed the $d'$ measure from signal detection theory (*Green and Swets, 1966*):

$$d'(\theta) = \frac{|\mu(\theta) - \mu(0)|}{\sqrt{(\sigma^2(\theta) + \sigma^2(0))/2}}$$

(9)

where $\mu(\theta)$ and $\sigma^2(\theta)$ are the mean and variance of the firing-rate distribution at head orientation $\theta$, and $\mu(0)$ and $\sigma^2(0)$ are the mean and variance of the firing-rate distribution at an angle of zero (i.e., vertical), respectively. The $d'$ values were then plotted as a function of $\theta$ for regular and irregular units.

## Spike-timing precision

To study the precision of spike timing in otolith afferents and to quantify the timescales at which these neurons operate to encode linear self-motion, we employed a classification method based on the Victor-Purpura (VP) measure (*Victor and Purpura, 1996*) as well as the van Rossum spike distance metric (VR) (*van Rossum, 2001*). First, to avoid non-stationarities in the response, we discarded the first and last 500 ms of each 10s-long epoch of broadband noise stimulus and split the remaining stimulus duration into nine 1s-long segments (i.e., nine different categories of self-motion stimuli). As mentioned earlier, for each neuron, we obtained the spike train during at least 6 (6–30) repetitions of these stimuli. For each category, one spike train was randomly chosen as a template and the remaining spike trains were assigned to one of the nine categories of stimulus based on the spike distance measure (see below). This procedure was repeated 30 times by drawing different template choices, and averages were then computed to construct a confusion matrix (*Figure 3—figure supplement 1*). The diagonal elements of this matrix correspond to the percentage of correctly classified spike trains (% correct), which was used as a measure of discrimination performance. Note that the chance level for classification accuracy was 11% because the probability that a spike train could be correctly assigned by chance in a discrimination procedure with nine classes is 1/9.

Distances between spike trains were estimated using the van Rossum (*van Rossum, 2001*) and the Victor-Purpura (*Victor and Purpura, 1996*) metrics. The van Rossum metric was computed in the following way. The spike train was first convolved with a decaying exponential kernel with time constant τ:

$$f(t) = \sum_{i=1}^{M} \Theta(t - t_i) e^{\frac{-(t-t_i)}{\tau}} \tag{10}$$

where $t_i$ is the $i^{th}$ spike time, M is the total number of spikes, and $\Theta(t)$ is the Heaviside step function ($\Theta(t) = 0$ if $t < 0$ and $\Theta(t) = 1$ if $t \geq 0$). The distance between two spike trains $R_j(t)$ and $R_k(t)$ was then defined as the Euclidean distance between their corresponding filtered traces, $f_{R_j}(t)$ and $f_{R_k}(t)$:

$$D_{vanRossum}\left(f_{R_j}, f_{R_k}\right)_\tau = \sqrt{\frac{1}{\tau} \int_0^\infty \left[f_{R_j}(t) - f_{R_k}(t)\right]^2} \tag{11}$$

Note that the parameter $\tau$ governs the temporal resolution of the metric; by varying $\tau$ ($1 \leq \tau \leq 2000$ ms) and repeating the classification procedure mentioned above, we investigated the impact of different timescales of the neuronal response on discrimination performance. When $\tau$ is small, the metric acts as a 'coincidence detector' because even minor differences in spike timing contribute to the distance, whereas at larger timescales the difference in total spike count matters and thus the metric becomes more of a 'rate difference counter' (**van Rossum, 2001**).

The Victor-Purpura metric is computed as the minimum total cost of transforming a spike train into another spike train through a series of basic operations. Specifically, insertion and deletion of a single spike have an associated cost of 1 each, whereas shifting a spike in time by an amount $\Delta t$ has an associated cost of $q\Delta t$ (**Victor and Purpura, 1997**; **Victor and Purpura, 1996**), where q (in units of 1/sec) is a parameter that determines the relative sensitivity of the metric to spike count and spike timing. Note that the quantity 1/q is a measure of temporal resolution in this metric and is related to the time constant $\tau$ in the van Rossum metric.

## Modeling

We built a leaky integrate-and-fire neuron to model the activity of otolith afferents using equations as follows:

$$C_m \frac{dV}{dt} = -gV(t) + I_{bias} + \sigma_{signal}S(t) + \sigma_{noise}\xi(t) \tag{12}$$

$$V(t) \geq \theta \rightarrow V(t^+) = 0$$

where $C_m$ is the membrane capacitance ($C_m$ = 1 nF), V(t) is the membrane potential, $g$ is the membrane conductance for the leak current ($g$ = 0.22 µS), $I_{bias}$ is a bias current (which is used to simulate the resting discharge of otolith afferents), $S(t)$ is the input current, which consisted of a broadband noise current (15 Hz cutoff) similar to the actual head acceleration stimuli applied to stimulate the afferents, and $\xi$ is a Gaussian white noise process with zero mean and standard deviations of $\sigma_{noise}$. To account for the known response dynamics of both regular and irregular otolith afferents, the stimulus $S(t)$ used in the model was obtained by filtering the original broadband noise current using the transfer functions of regular and irregular units as in previous studies (**Jamali et al., 2016**; **Schneider et al., 2015**). The parameters $\sigma_{noise}$ and $\sigma_{signal}$ determine the response variability and the strength of the signal, respectively. When *V(t)* is greater than or equal to the threshold $\theta$ (i.e., –50 mV), *V(t)* is immediately reset to 0 mV and a spike is said to have occurred at time t. *Equation (11)* was numerically integrated using an Euler-Maruyama algorithm with a time step of 0.025 ms. The spiking responses from the model were analyzed in the same way as the experimental data.

For the regular model neuron, parameter values were: $I_{bias}$ = 3.53 nA, $\sigma_{noise}$ = 0.14 nA, $\sigma_{signal}$ = 0.14 nA. For the irregular model neuron, parameter values were: $I_{bias}$ = 3.53 nA, $\sigma_{noise}$ = 1.9 nA, $\sigma_{signal}$ = 1.9 nA. Parameter values were set such that the responses of the regular and irregular model neurons mimicked experimental data (resting discharge; regular model, CV*=0.06; irregular model, CV*=0.42). We co-varied both sensitivity (i.e., $\sigma_{signal}$) and variability (i.e., $\sigma_{noise}$) in our model keeping their ratio constant and computed the information transmitted by firing rate and spike timing as described above for the experimental data.

## Statistics

Our sample sizes were similar to those generally employed in the field (*Si et al., 1997*; *Yu et al., 2012*; *Jamali et al., 2013*; *Yu et al., 2015*; *Laurens et al., 2017*). All values are expressed as mean ± SEM throughout. Statistical significance was determined by using Wilcoxon ranksum tests at the p=0.05 level.

## Acknowledgements

The authors would like to thank Soroush Sadeghi for his contribution to data collection and S Nuara and W Kucharski for excellent technical assistance. This study was supported by the Canadian Institutes for Health Research and the Canada research chairs.

## Additional information

### Funding

| Funder | Grant reference number | Author |
|---|---|---|
| Canadian Institutes of Health Research | | Kathleen E Cullen Maurice J Chacron |
| Canada Research Chairs | | Maurice J Chacron |
| National Institutes of Health | DC2390 | Kathleen E Cullen |

The funders had no role in study design, data collection and interpretation, or the decision to submit the work for publication.

### Author contributions

Mohsen Jamali, Conceptualization, Data curation, Software, Formal analysis, Validation, Investigation, Visualization, Methodology, Writing—original draft, Writing—review and editing; Jerome Carriot, Data curation, Investigation, Methodology; Maurice J Chacron, Conceptualization, Supervision, Investigation, Methodology, Writing—original draft, Writing—review and editing; Kathleen E Cullen, Conceptualization, Resources, Software, Supervision, Funding acquisition, Investigation, Methodology, Writing—original draft, Project administration, Writing—review and editing

### Author ORCIDs

Mohsen Jamali  https://orcid.org/0000-0002-1750-7591
Maurice J Chacron  http://orcid.org/0000-0002-3032-452X
Kathleen E Cullen  https://orcid.org/0000-0002-9348-0933

### Ethics

Animal experimentation: All experimental protocols were approved by the McGill University Animal Care Committee (#2001-4096) and were in compliance with the guidelines of the Canadian Council on Animal Care. Two male macaque monkeys (Macaca fascicularis) were prepared for chronic extracellular recording under aseptic conditions. The surgical preparation was similar to that previously described (Dale & Cullen, 2013). Animals (aged 6 and 8 years old) were housed in pairs on a 12 hour light/dark cycle.

### Decision letter and Author response

Decision letter https://doi.org/10.7554/eLife.45573.015
Author response https://doi.org/10.7554/eLife.45573.016

## Additional files

### Supplementary files
• Transparent reporting form
DOI: https://doi.org/10.7554/eLife.45573.011

### Data availability
All data generated or analyzed during this study are included in the manuscript and supporting files. Source data and Matlab codes have been deposited on Figshare under the URL: https://doi.org/10.6084/m9.figshare.8251613.v1.

The following dataset was generated:

| Author(s) | Year | Dataset title | Dataset URL | Database and Identifier |
|---|---|---|---|---|
| Jamali M, Carriot J, Chacron MJ, Cullen KE | 2019 | Coding strategies in the otolith system differ for translational head motion vs static orientation relative to gravity | https://dx.doi.org/10.6084/m9.figshare.8251613.v1 | Figshare, 10.6084/m9.figshare.8251613 |

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
