## [Decision Letter]

[Editors’ note: this article was originally rejected after discussions between the reviewers, but the authors were invited to resubmit after an appeal against the decision.]

Thank you for submitting your work entitled "Channel specific coding in the otolith system yields robust estimates of head motion and orientation relative to gravity" for consideration by *eLife*. Your article has been reviewed by three peer reviewers, and the evaluation has been overseen by a Senior Editor.

Our decision has been reached after consultation between the reviewers. Based on these discussions and the individual reviews below, we feel that the main point in the current manuscript about selective coding of translation and tilt stimulus is interesting and important, but unfortunately it does not get sufficient support from the current data with the current experimental design. Thus, we regret to inform you that your work will not be considered further for publication in *eLife*.

*Reviewer #1:*

In the current study, Jamali et al., explored information coding in two subtypes of vestibular otolith afferents under two stimulus conditions: translation of the head/whole body, and static tilt of the head/whole body. The main finding is that the irregular otolith afferents show larger sensitivity to body translation compared to the regular subtypes, due to their distinguished response dynamics embedded in each class. In contrast, this response sensitivity is reversed under the static tilt (relative to gravity) condition, that is, the regular otolith afferents become more sensitive. These results, particularly the responses of the two types of afferents under static tilt condition, is new and worth reporting. However, there are also concerns here:

1) The current data clearly show that the irregular afferents are more sensitive, or carry more information about translation, including both approximate-natural or sinusoidal translation, compared to the regular counterparts. However, this does not necessarily exclude the possibility that the regular class can also encode translation information. Similarly, to static tilt, the irregular afferents are less sensitive compared to the regular counterparts, but again this does not mean that they cannot encode static tilt. Thus the authors' conclusion about irregular and regular respectively dealing with translation and static tilt, lacks sufficient and direct evidence support.

2) More importantly, the authors further implied that the afferents itself can solve the Einstein's Equivalence Principle problem, through the irregular and regular channels. This is wrong. First, it is well known that due to the physical properties, the otolith and its afferents have no way to solve this problem (e.g. Fernandez and Goldberg, 1976). Second, a series of previous studies (e.g. Angelaki and Dickman, 1999; Angelaki et al., 2004; Liu et al., 2011; Yukushiva et al., 2007) have designed delicate experimental paradigm to directly examine whether different brain stages, from the afferents to cerebellum and cerebral cortex, could solve the translation-tilt ambiguity problem. Importantly, this is done by providing sinusoidal translation, and a "matched" tilt stimulus. The key point is to use a "dynamic" tilt stimulus that provides momentary matched acceleration signal that can be perfectly cancelled or superimposed with the translation stimulus, allowing to examine whether the target neuron really prefers translation or tilt. Using this paradigm, previous researchers have found that otolith afferents respond equally under translation and its matched tilt stimulus. It would be nice for the current work, the authors could further examine how the two types of afferents respond under these stimulus condition, and thus can push this filed a step forward. Unfortunately, the authors here adopt static tilt, providing totally unmatched stimulus compared to the translation (either frequency varied or fixed sinusoidal), and thus cannot directly address whether each type of afferents selectively encode translation against (matched) tilt stimulus. In fact, if the authors have performed the correct experiment paradigm using matched dynamic tilt and translation stimulus, they are very likely to find that their results would be reversed, that is, irregular afferents would now become more sensitive to the (dynamic) tilt relative to gravity compared to the regular counterpart.

3) Thus, there is a fundamental difference between using the dynamic and static tilt stimulus. The current results cannot be used to address the classical Einstein's Equivalence Principle problem. To avoid confusions with the general audience, the author should use correct terms in their text. However, so far there is only one place which is in the last sentence in the Introduction part, the authors have used the word of "static", but did not do so for the other key places including the Abstract, Result and Discussion.

4) Even under the static tilt stimulus as used in the current study, the authors did not show clearly about how they provide the stimulus and compute the responses. How fast were the animals rotated from upright to the expected tilted position? What time window (suppose after tilted) was used to compute firing rate to get the d' value? The authors need to present the whole temporal dynamics of their data to show how they do this, as for their translation data.

5) The difference in the firing statistics and coding ability to the translation stimulus between the irregular and regular otolith afferents are well illustrated in the text, from Figure 1-4. However, these results were sort of reported and overlapped in previous studies, for example, in the authors' own group (Jamali et al., 2013, Figure 2). The only significant difference I see here is that they tested the responses to translation under a frequency-varied condition (what they call "natural"). However, this result is sort of expected from their previous results under the frequency-fixed sinusoidal translation condition, and thus may not be novel enough for the journal. Can the authors clearly state the significant progress they made about the irregular and regular afferents coding ability compared to previous results (e.g. Jamali et al., 2013)?

*Reviewer #2:*

In this manuscript, Jamali and colleagues analyze the information transmitted by regular and irregular otolith afferents of macaque monkeys, recorded during low-frequency tilt and high-frequency translation. They demonstrate that irregular afferent encode high-frequency otolith stimulation better than regular afferent, due to higher response gains and phase locking. In contrast, regular afferent encode static (i.e. low-frequency) otolith stimulus better.

In line with earlier works, Jamali's analysis of afferent spiking is comprehensive, state-of-the art and well presented. The dataset shown here is also impressive, considering the difficulty of recording otolith afferents. Altogether, this study offers a novel and highly informative study of otolith firing, in particular in the high-frequency range.

Unfortunately, the authors draw an incorrect and flawed conclusion from these results, which is that irregular afferent preferentially encode translational motion whereas regular afferent preferentially encode head orientation relative to gravity. Based on Einstein's equivalence principle, linear and gravitational acceleration are physically indistinguishable, and all otolith afferents respond identically to tilt and translations. The author's implied logic is that natural tilt and translation are segregated in the frequency domain, tilt being confined to low frequencies and translations high frequencies. However, the authors make no effort to justify this assumption. Actually, their own studies (Carriot et al. 2014, 2017) show that natural tilt movements extend well into the high frequency range (>1Hz) where irregular afferent respond preferentially. Furthermore, it is accepted that low-frequency translations are infrequent, such that low-frequency otolith stimuli can be interpreted as tilt. However, regular afferents are also sensitive to mid- or high-frequency stimuli, since they have a flat gain curve. Therefore, they can pick up translations. Thus, both types of afferents are expected to carry mixture of tilt and translation signals. None of this is taken into consideration by the authors, who simply restrict their stimuli to low-frequency tilt and high-frequency translation, and therefore wrongly equate low frequency and tilt, and high frequency and translation. Thus, their conclusion that regular/irregular afferents are 'parallel channels that preferentially encode translations and gravity' is simply biased by their choice of stimuli.

The authors also fail to position their study relative to other works in the field. Until the late 90's, there have been two hypotheses about how the brain discriminates tilt from translation. The 'frequency segregation' hypothesis (Paige, Tomko, Telford, Raphan, Cohen…) posits that the brain segregates tilt and translation by frequencies, i.e. interprets low/high frequency otolith stimuli as tilt/translations respectively (and indeed Raphan 2002 suggests that this could be accomplished by weighting regular/irregular otolith inputs). The 'internal model' hypothesis, advanced e.g. by Mayne, Young and Oman, Merfeld and recently Laurens, Karmali, posits that central brain areas merge semi-circular canal and otolith information to separate tilt from translation (this hypothesis also explains that low-frequency otolith stimulation is preferentially interpreted as tilt based on the statistics of natural stimuli). The frequency segregation hypothesis predicts that the brain can't discriminate tilt and translation stimuli if they have identical frequencies. This has been refuted directly multiple times (in particular by Angelaki's group) and the internal model hypothesis has been repetitively validated by neuronal recordings (e.g. Angelaki, Laurens), psychophysics (e.g. Merfeld's group, Hess, McNeilage) and modeling (e.g. Laurens, Karmali, Bos and Bless). Although the authors are well aware of these progresses, they chose to ignore them (citing only one paper from the last 20 years!) and instead essentially resurrect the frequency segregation hypothesis.

This study contains new and important results on how otoliths transmit low and high frequency motion. I would encourage the authors to focus on this. Should they wish to maintain their hypothesis that regular/irregular afferents preferentially encode tilt and translation respectively, they should at least support it by a sound analysis of the statistics of natural stimuli, and place it in the context of current knowledge on tilt/translation discrimination, based on an accurate review of the literature.

To recapitulate:

Abstract fourth and final sentences; subsection “Parallel processing of gravito‐inertial forces: implications for perception” first sentence in first paragraph and final sentence in the second paragaph: These statements are simply false. Neither regular nor irregular afferent transmit tilt or translation preferentially, unless one assumes that tilt and translation occur in distinct frequency ranges. Such assumptions are never explicitly made, quantified or justified in the manuscript. Even if it was the case, each of this statement should be reworded to indicate that they are valid only under this assumption. Otherwise, they could actually mislead the reader to think that otoliths can actually resolve the tilt/translation ambiguity, which is physically impossible.

Subsection “Parallel processing of gravito‐inertial forces: implications for perception”: This part is extremely incomplete and misses all the modern literature (i.e. post year 2000) on tilt/translation discrimination. The authors should discuss the fact that neurons in the vestibulo-cerebellum can distinguish tilt and translation stimuli that have identical frequencies and magnitudes, which is physically impossible to accomplish in the otoliths, and therefore demonstrates the existence of a central process to distinguish tilt from translation.

*Reviewer #3:*

The authors investigated the type of information encoded by regular and irregular otolith afferents in monkeys during "naturalistic self-motion stimulation" They used appropriate analytical methods. Data analysis suggests that irregular otolith afferents use timing and rate code and are better suited to detect changes in "gravitionertial acceleration" Regular otolith afferents use only rate code and are better suited at estimating "gravitoinertial acceleration" during the steady state (such as static head tilt). Remarkably, their finding may reconcile two "alternative" hypotheses: that the brain uses only otolith‐related signals to generate a prediction of tilt and translation, vs that the brain uses a combination of otolith and canal signal. Likely, both strategies are used by the brain for spatial navigation. The authors also used computer simulations that indicate that the large sensitivity and variability of irregular afferents can explain the functional differences between afferent classes. The major finding of this study, according to the authors, is that irregular and regular afferent represent two information channels. One channel, regular afferents, relays mostly head orientation signal to the CNS. A second channel, irregular afferents, relay mostly translation. I believe, however, that the interpretation of the data is misleading or incorrect, as both regular and irregular afferents detect combined gravitational and translational information, but perhaps at different optimal frequency bands.

The major issue with this manuscript is that the conclusion, which is the major selling point of the manuscript, is misleading. The authors hypothesize that "Regular afferents preferentially transmit information about head orientation while irregular afferents instead preferentially transmit information about translational self‐motion" If true, this would be very relevant for the field because it would provide a novel solution to the ambiguity found in the response of otolith afferents (which carry gravito-inertial information). However, this hypothesis was not tested directly or indirectly in this study. Thus, it is misleading and should be rewritten through the manuscript to represent the data faithfully, starting with changing the title. There are many ways to test the authors' hypothesis. Perhaps, the simplest way would be to compare the response of afferents during translation and tilt stimuli that contain the same dynamics (i.e., stimuli that generate the same gravitoinertial acceleration). What the data show is that regular and irregular afferents show complementary frequency bands to discriminate gravitoinertial information.

Data analysis suggests that irregular otolith afferents use timing and rate code, while regular otolith afferents use only rate code. This is an important finding because it is opposite to the coding found in the semicircular canal afferents by the same group (Sadeghi et al., 2007). What are the implications that canal and otolith afferent show different coding strategies? Could these differences be due to differences in stimuli (e.g., frequency) or analytical methods between this experiment and that of Sadeghi et al., 2007?

It would help to have model simulation results in the last part of the result section (Figure 6).

[Editors’ note: what now follows is the decision letter after the authors submitted for further consideration.]

Thank you for choosing to send your work entitled "Irregular and regular otolith afferents differentially encode naturalistic translational head motion and static orientation relative to gravity" for consideration at *eLife*. Your letter of appeal has been considered by a Senior Editor and the original reviewers, and we are prepared to consider a revised submission with no guarantees of acceptance.

Included below are comments from the reviewers, who were asked to comment on your appeal and revised manuscript. I am including them all in full because I think that they will all be useful to you for making revisions. Reviewer #3 suggested additional experiments, but after discussion we concluded that although such experiments would be quite informative, they are not necessary for a revision for *eLife*.

*Reviewer #1:*

In the current revised version of the manuscript, it is good to see that the authors have made significant improvements in their description and interpretation about their findings in that irregular and regular afferents are sensitive under different stimulus dynamics. However, some important citations are still missing, and there are still a few places that need to be further improved for publication:

1) As indicated in my comments in the first run, the findings about different neuronal sensitivity for the two types of afferents under high frequency translation and static tilt stimulus do not necessarily imply that the brain is using the strategy as expected by us, or the authors. First of all, irregular afferents are not unresponsive at all in the static tilt case, and the same logic applies to the regular afferents under the dynamic translation stimuli. Second, whether information about dynamic translation and static tilt stimulus is really decoded from the irregular and regular afferents, respectively, remains unknown under lacking of further experiments like the causality manipulation. Thus, the current statement in the Abstract was made to be too strong. Something like this is better: "Together, our results indicate that irregular and regular otolith afferents may use different strategies to encode naturalistic self-14 motion and static head orientation relative to gravity."

2) Subsection “Differential encoding of gravito-inertial forces by irregular and regular otolith afferents: implications for perception” second paragraph: About monkey's discriminability of self-motion direction, Gu and Angelaki, 2007, should be added in addition to Yu and Angelaki, 2015. About the relatively higher neuronal threshold of the otolith afferents compared to the behavior, Yu and Angelaki's work (2015) should be added in addition to Jamali 2013's work.

3) Third paragraph of subsection “Differential encoding of gravito-inertial forces by irregular and regular otolith afferents: implications for perception”: A comparison with the previous experiments, in particular, the difference in the methodology between the current study and the previous ones needs to be further expanded and clarified. As indicated in the first run of the reviewing process, a series of important works (see below) used a well established method by providing matched translation and dynamic tilt stimuli to examine and compare how neurons would respond under these two conditions with identical acceleration. These works are currently missing and should be included:

a) Angelaki et al., 2004, Neurons compute internal models of the physical laws of motion

b) Liu and Angelaki, 2011, Response Dynamics and Tilt versus Translation Discrimination in Parietoinsular Vestibular Cortex

c) Laurens and Angelaki, 2013, Neural representation of orientation relative to gravity in the macaque cerebellum

*Reviewer #2:*

In my initial review, I have appreciated the quality of the analyses presented in this manuscript, but I had major concerns regarding the study's overall message that regular and irregular afferents preferentially encode tilt and translation, respectively. However, I am glad to re-consider this study, provided that these concerns are addressed. I find that the authors have indeed addressed them, but not completely, and I would encourage them to revise their manuscript further.

Major points:

The major issue, expressed independently by all three original reviewers, is that the manuscript concluded that irregular and regular afferents preferentially encode head translation and tilt, respectively. I am somewhat bewildered by the author's argument that they never meant to imply this (see e.g. 384-388 of their initial manuscript), but I will let it slide. All three original reviewers independently understood that the manuscript made this conclusion, and there is no doubt that the vast majority of readers would have understood it too. One way or another, the manuscript needed to be revised.

In the present version, the authors have largely clarified this point. However, I think that some readers would still get the wrong message and that additional clarifications are needed:

– Results second paragraph: when they first describe their protocol, the authors should explain that it is representative of naturalistic medium/high-frequency translation or tilt, equivalently (and that their conclusions are equally valid for these types of motion).

– They should generally refer this protocol as a "naturalistic self-motion protocol" and not to "naturalistic translation" (as they already do in many instances). For instance, they should change subsection “Contributions of spike timing towards the encoding of translational self-motion by otolith afferents”; “Increases in variability and sensitivity lead to greater information transmission and spike timing precision”; “Irregular otolith afferents display phase-locking to sinusoidal self-motion translational stimuli.”; “Regular otolith afferents better encode differences in static head orientation relative to gravity than their irregular counterparts.” and Discussion, first paragraph.

In conclusion for this point, the authors' manuscript is very informative about how regular and irregular afferents differentially sense naturalistic, mid/high frequencies tilt and translation, e.g. as experienced when moving naturally. It should be made perfectly clear throughout the entire manuscript that this applies to tilt and translation. Regarding low-frequency motion, it is fine to propose that regular afferent may sense preferentially "quasi-static" stimuli, which, under natural circumstances, would mean head tilt.

Another major point, which I raised in the initial review, is that the discussion of how the brain discriminates tilt from translation is extremely incomplete. This has not been improved. Stating that the brain can "theoretically" distinguish tilt and translation by integrating otoliths and canal afferents was fine in 1998, but it is a gross misrepresentation of the current knowledge in the field. It has been demonstrated over and over that the brain discriminates tilt from translation centrally, and the underlying neuronal bases have been largely explored (predominantly Dora Angelaki's group). Therefore, the authors have to provide and short and up-to-date summary of the current knowledge on this topic.

By the way, in paragraph three of subsection “Differential encoding of gravito-inertial forces by irregular and regular otolith afferents: implications for perception”, the authors state that further studies should investigate how the brain processes otoliths afferent to give rise to perception during low frequency motion. They will be glad to learn that this has already been studied. Neuronal correlates of the somatogravic effect, where low-frequency translations are interpreted as tilt, have been identified by Laurens et al., 2013, in the vestibulo-cerebellum. And the cerebellum contributes to self-motion perception (Dahlem et al., 2016).

*Reviewer #3:*

In this manuscript, Jamali and colleagues show empirical and modeling data supporting the hypothesis that irregular and regular afferents use different coding methods suitable for representing high-frequency and low-frequency gravitoinertial acceleration, respectively. The manuscript has improved significantly from the original version by departing from its previous claim that the tilt/translation ambiguity is solved at the afferent level.

The general strengths of the manuscript remain the same as in the previous version and include: a set of technically challenging experiments, well suited analytical tools and statistics, and the use of modeling work that can explain the results. It remains controversial whether, or to a which degree, the CNS is decoding information from vestibular afferents as proposed here since both regular and irregular afferents inform about translation and static tilt. Thus, the manuscript may not be of sufficient impact to merit publication in this journal. Interestingly, the authors' hypothesis produces testable predictions on the effect of silencing irregular afferents (i.e., using anodal currents) in perceptual thresholds. Addition of these experiments would raise the impact of the manuscript to merit publication in *eLife*.

---

## [Author Response]

[Editors’ note: the author responses to the first round of peer review follow.]

First, we would like to thank the editors and reviewers for their efforts in reviewing our paper and providing feedback. We note that there is a consensus amongst all 3 reviewers that our study provides a new and significant contribution to the neuroscientific community and is hence worth reporting. Specifically:

Reviewer #1. “These results, particularly the responses of the two types of afferents under static tilt condition, are new and worth reporting”.

Reviewer #2. “This study contains new and important results on how otoliths transmit low and high frequency motion” and “In line with earlier works, Jamali's analysis of afferent spiking is comprehensive, state-of-the art and well presented. The dataset shown here is also impressive, considering the difficulty of recording otolith afferents. Altogether, this study offers a novel and highly informative study of otolith firing, in particular in the high-frequency range.”

Reviewer #3. “Remarkably, their finding may reconcile two "alternative" hypotheses: that the brain uses only otolith‐related signals to generate a prediction of tilt and translation, vs that the brain uses a combination of otolith and canal signal. Likely, both strategies are used by the brain for spatial navigation. What the data show is that regular and irregular afferents show complementary frequency bands to discriminate gravitoinertial information….This is an important finding because it is opposite to the coding found in the semicircular canal afferents by the same group (Sadeghi et al.,

2007).”

Thus, overall, the reviewers acknowledged that our results and analysis are not only rigorous and robust, but also make significant contribution to the literature.

Indeed, our study is the first to have tested the static and dynamic responses of otolith afferents over a broad physiological range up to 15 Hz using naturalistic stimuli. Based on our data and analyses, our findings make numerous novel and important contributions to the literature, including:

– We establish for the first time, the coherence, information content, and coding strategies of otolith afferents during naturalistic stimuli.

– We show for the first time that irregular and regular primate otolith afferents use different strategies to encode naturalistic translational head motion and static orientation relative to gravity.

– We reproduce these results using an integrate and fire model and explore how the interplay between variability and sensitivity influence the coding strategies employed by otolith afferents.

– We predict and then confirm that otolith afferents actually phase lock for sinusoidal stimulation applied over the physiologically relevant frequency range.

These findings are all novel and highly informative for both the vestibular field as well as the neuroscience community in general.

Based on our reading of the editor’s letter and reviews, we believe that the paper was not rejected due to any issues relating to the validity of any of the findings listed above. Instead, it appears that the paper was rejected due to a misunderstanding of our conclusions, due in large part to a single paragraph in the discussion. In particular, the main concerns of reviewers 1 and 2 is that they felt we implied that otolith afferents alone can solve the ambiguity between tilt and translation (i.e., “Einstein's Equivalence Principle problem”). Specifically,

Reviewer #1 “More importantly, the authors further implied that the afferents itself can solve Einstein's Equivalence Principle problem, through the irregular and regular channels..”

Reviewer #2 “Based on Einstein's equivalence principle, linear and gravitational acceleration are physically indistinguishable, and all otolith afferents respond identically to tilt and translations. The author's implied logic is that natural tilt and translation are segregated in the frequency domain, tilt being confined to low frequencies and translations high frequencies.”

However, we never stated or intended to imply that afferents alone can solve the Einstein's Equivalence Principle problem. Our goal in this section of the discussion was simply to review and acknowledge the literature on this topic. In fact, our argument was quite the opposite, and in our original submission, we actually had explicitly stated that “As mentioned above, Einstein’s equivalence principle posits that the forces resulting from both are physically indistinguishable from one another …Thus, otolith afferents transmit ambiguous information to the brain during everyday life”, as they encode both tilt and translation.

Overall, we believe there was a misunderstanding regarding our use of the term “head orientation” in the text that followed. Throughout the text we used the term “head orientation” to refer to “static head orientation” and not changes in head orientation or dynamic tilt. Since, in our original submission, we had always used the term “static head orientation” to introduce results and discussion related to this aspect of our study, we did not think it was necessary to reiterate the word “static” each time since it made the writing seem overly repetitive. However, we now appreciate that this has led the reviewers to misunderstand our interpretation of the results.

Based on these comments, which only required only text editing changes we have revised the manuscript. For example, we now consistently precede any reference to the coding of head orientation with the word static in the revised manuscript.

Furthermore, we have revised the Discussion to more clearly state that otolith afferents alone cannot solve Einstein's Equivalence Principle. Specifically: “Our results show that regular and irregular otolith afferents use different coding strategies in order to provide estimates of both dynamic head motion and static orientation relative to gravity..” and now more clearly make our point that future work is needed to understand how information carried by regular and irregular otolith afferents during natural stimulation gives rise to perception. Additionally, we have revised our title to: “Irregular and regular primate otolith afferents differentially encode naturalistic translational head motion and static orientation relative to gravity.”

Finally, as detailed above and acknowledged by the reviewers, as well as considering the difficulty of recording from otolith afferents, we again emphasize that our study provides significant novel information about the response properties of these neurons.

We strongly believe that our results constitute a major conceptual advance with respect to previous studies, and has high impact not only for the vestibular field but also for those interested in neural coding and a broader group of neuroscientists interested in how the brain processes sensory inputs to ensure accurate behavior.

[Editors’ note: the author responses to the re-review follow.]

Included below are comments from the reviewers, who were asked to comment on your appeal and revised manuscript. I am including them all in full because I think that they will all be useful to you for making revisions. Reviewer #3 suggested additional experiments, but after discussion we concluded that although such experiments would be quite informative, they are not necessary for a revision for eLife.

Reviewer #1:

In the current revised version of the manuscript, it is good to see that the authors have made significant improvements in their description and interpretation about their findings in that irregular and regular afferents are sensitive under different stimulus dynamics. However, some important citations are still missing, and there are still a few places that need to be further improved for publication:1) As indicated in my comments in the first run, the findings about different neuronal sensitivity for the two types of afferents under high frequency translation and static tilt stimulus do not necessarily imply that the brain is using the strategy as expected by us, or the authors. First of all, irregular afferents are not unresponsive at all in the static tilt case, and the same logic applies to the regular afferents under the dynamic translation stimuli. Second, whether information about dynamic translation and static tilt stimulus is really decoded from the irregular and regular afferents, respectively, remains unknown under lacking of further experiments like the causality manipulation. Thus, the current statement in the Abstract was made to be too strong. Something like this is better: "Together, our results indicate that irregular and regular otolith afferents may use different strategies to encode naturalistic self-14 motion and static head orientation relative to gravity."

We have changed the Abstract as the reviewer suggested.

2) Subsection “Differential encoding of gravito-inertial forces by irregular and regular otolith afferents: implications for perception” second paragraph: About monkey's discriminability of self-motion direction, Gu and Angelaki, 2007, should be added in addition to Yu and Angelaki, 2015. About the relatively higher neuronal threshold of the otolith afferents compared to the behavior, Yu and Angelaki's work (2015) should be added in addition to Jamali 2013's work.

The references were added per reviewer’s suggestion.

3) Third paragraph of subsection “Differential encoding of gravito-inertial forces by irregular and regular otolith afferents: implications for perception”: A comparison with the previous experiments, in particular, the difference in the methodology between the current study and the previous ones needs to be further expanded and clarified. As indicated in the first run of the reviewing process, a series of important works (see below) used a well-established method by providing matched translation and dynamic tilt stimuli to examine and compare how neurons would respond under these two conditions with identical acceleration. These works are currently missing and should be included:a) Angelaki et al., 2004, Neurons compute internal models of the physical laws of motionb) Liu and Angelaki, 2011, Response Dynamics and Tilt versus Translation Discrimination in Parietoinsular Vestibular Cortexc) Laurens and Angelaki, 2013, Neural representation of orientation relative to gravity in the macaque cerebellum

We appreciate reviewer’s concern and have revised the Discussion to provide description of the neural correlates of how the brain can distinguish between tilt and translation and have included the suggested references.

Reviewer #2:In my initial review, I have appreciated the quality of the analyses presented in this manuscript, but I had major concerns regarding the study's overall message that regular and irregular afferents preferentially encode tilt and translation, respectively. However, I am glad to re-consider this study, provided that these concerns are addressed. I find that the authors have indeed addressed them, but not completely, and I would encourage them to revise their manuscript further.

We thank the reviewer for his/her support and for providing constructive feedback that helped us to improve the quality of the paper. We appreciate the reviewer’s concerns and have further revised the manuscript to clarify this point.

Major points:The major issue, expressed independently by all three original reviewers, is that the manuscript concluded that irregular and regular afferents preferentially encode head translation and tilt, respectively. I am somewhat bewildered by the author's argument that they never meant to imply this (see e.g. 384-388 of their initial manuscript), but I will let it slide. All three original reviewers independently understood that the manuscript made this conclusion, and there is no doubt that the vast majority of readers would have understood it too. One way or another, the manuscript needed to be revised.In the present version, the authors have largely clarified this point. However, I think that some readers would still get the wrong message and that additional clarifications are needed:

We have revised the manuscript to improve the clarity of our main conclusions. We have further revised the text to clarify this point throughout the manuscript.

– Results second paragraph: when they first describe their protocol, the authors should explain that it is representative of naturalistic medium/high-frequency translation or tilt, equivalently (and that their conclusions are equally valid for these types of motion).– They should generally refer this protocol as a "naturalistic self-motion protocol" and not to "naturalistic translation" (as they already do in many instances). For instance, they should change subsection “Contributions of spike timing towards the encoding of translational self-motion by otolith afferents”; “Increases in variability and sensitivity lead to greater information transmission and spike timing precision”; “Irregular otolith afferents display phase-locking to sinusoidal self-motion translational stimuli.”; “Regular otolith afferents better encode differences in static head orientation relative to gravity than their irregular counterparts.” and Discussion, first paragraph.

We appreciate the reviewer’s comment. We have revised the manuscript to improve clarity regarding the actual experimental protocols that were used and the implications of our results. Specifically, we now describe our stimulation protocol in detail at the beginning of the results by clearly defining the different stimuli used and now consistently refer to these using the same denominations throughout the manuscript. We have also expanded the discussion to clearly state the implications of our results: “Specifically, we predict that irregular otolith afferents will outperform their regular counterparts at encoding dynamic changes in head orientation, provided that their temporal frequency content is high enough (i.e., > 0.1 Hz). Moreover, we predict that regular otolith afferents will outperform their irregular counterparts at encoding low frequency (i.e., lower than 0.1 Hz) translational self-motion.”

In conclusion for this point, the authors' manuscript is very informative about how regular and irregular afferents differentially sense naturalistic, mid/high frequencies tilt and translation, e.g. as experienced when moving naturally. It should be made perfectly clear throughout the entire manuscript that this applies to tilt and translation. Regarding low-frequency motion, it is fine to propose that regular afferent may sense preferentially "quasi-static" stimuli, which, under natural circumstances, would mean head tilt.

We now mention in the text that, while our results showing that irregular otolith afferents outperform their regular counterparts for high (i.e., > 0.1 Hz) frequencies were obtained using translational self-motion stimuli, we predict that they will also apply to dynamic changes in head orientation provided that these also contain high (i.e., > 0.1 Hz) frequencies. We also mention in the text, while our results showing that regular otolith afferents outperform their irregular counterparts at distinguishing between different head orientations relative to gravity were obtained using static head orientations, we predict that they will also apply to translational self-motion provided that the frequency content is low enough (i.e. < 0.1 Hz).

Another major point, which I raised in the initial review, is that the discussion of how the brain discriminates tilt from translation is extremely incomplete. This has not been improved. Stating that the brain can "theoretically" distinguish tilt and translation by integrating otoliths and canal afferents was fine in 1998, but it is a gross misrepresentation of the current knowledge in the field. It has been demonstrated over and over that the brain discriminates tilt from translation centrally, and the underlying neuronal bases have been largely explored (predominantly Dora Angelaki's group). I can perfectly understand that (one of) the authors feel(s) in conflict with Dora Angelaki; but this is not a justification for writing a biased discussion in turn. Whether the authors like it or not, their study will raise the question of how the brain distinguishes tilt from translation. Therefore, they have to provide and short and up-to-date summary of the current knowledge on this topic.

We have revised the Discussion to provide a short and up-to-date summary of the current knowledge on the topic of distinguishing between tilt and translation.

By the way, in paragraph three of subsection “Differential encoding of gravito-inertial forces by irregular and regular otolith afferents: implications for perception”, the authors state that further studies should investigate how the brain processes otoliths afferent to give rise to perception during low frequency motion. They will be glad to learn that this has already been studied. Neuronal correlates of the somatogravic effect, where low-frequency translations are interpreted as tilt, have been identified by Laurens et al., 2013, in the vestibulo-cerebellum. And the cerebellum contributes to self-motion perception (Dahlem et al., 2016).

We have revised the Discussion to include these studies.

Reviewer #3:

In this manuscript, Jamali and colleagues show empirical and modeling data supporting the hypothesis that irregular and regular afferents use different coding methods suitable for representing high-frequency and low-frequency gravitoinertial acceleration, respectively. The manuscript has improved significantly from the original version by departing from its previous claim that the tilt/translation ambiguity is solved at the afferent level.

We thank the reviewer for the positive feedback.

The general strengths of the manuscript remain the same as in the previous version and include: a set of technically challenging experiments, well suited analytical tools and statistics, and the use of modeling work that can explain the results. It remains controversial whether, or to a which degree, the CNS is decoding information from vestibular afferents as proposed here since both regular and irregular afferents inform about translation and static tilt. Thus, the manuscript may not be of sufficient impact to merit publication in this journal. Interestingly, the authors' hypothesis produces testable predictions on the effect of silencing irregular afferents (i.e., using anodal currents) in perceptual thresholds. Addition of these experiments would raise the impact of the manuscript to merit publication in eLife.

We agree with the reviewer that such experiments would be quite informative. However, as noted by the editor, these are beyond this scope of this study and have revised the Discussion to point out that future studies should test these predictions.